# Cybercrime Risk Found in Employee Behavior Big Data Using Semi-Supervised Machine Learning with Personality Theories

Kenneth David Strang [1,2,3]

[1]  Business Analytics Department, W3-Research, St. Thomas District, USVI 00802, USA;
    professor@kennethstrang.com
[2]  Plaster Graduate School of Business, University of the Cumberlands, Williamsburg, KY 40769, USA
[3]  Business | Supply Chain Management Department, State University of New York, Albany, NY 12246, USA

**Abstract:** A critical worldwide problem is that ransomware cyberattacks can be costly to organizations. Moreover, accidental employee cybercrime risk can be challenging to prevent, even by leveraging advanced computer science techniques. This exploratory project used a novel cognitive computing design with detailed explanations of the action-research case-study methodology and customized machine learning (ML) techniques, supplemented by a workflow diagram. The ML techniques included language preprocessing, normalization, tokenization, keyword association analytics, learning tree analysis, credibility/reliability/validity checks, heatmaps, and scatter plots. The author analyzed over 8 GB of employee behavior big data from a multinational Fintech company global intranet. The five-factor personality theory (FFPT) from the psychology discipline was integrated into semi-supervised ML to classify retrospective employee behavior and then identify cybercrime risk. Higher levels of employee neuroticism were associated with a greater organizational cybercrime risk, corroborating the findings in empirical publications. In stark contrast to the literature, an openness to new experiences was inversely related to cybercrime risk. The other FFPT factors, conscientiousness, agreeableness, and extroversion, had no informative association with cybercrime risk. This study introduced an interdisciplinary paradigm shift for big data cognitive computing by illustrating how to integrate a proven scientific construct into ML—personality theory from the psychology discipline—to analyze human behavior using a retrospective big data collection approach that was asserted to be more efficient, reliable, and valid as compared to traditional methods like surveys or interviews.

**Keywords:** employee behavior; retrospective big data; five-factor personality theory; cybercrime risk; cybersecurity; semi-supervised machine learning; action research; case study; fintech

## 1. Introduction

Cybercrime risk is a crucial global business topic because cyberattacks have resulted in significant financial losses [1–4]. Scholars have called for more scientific studies of cybercrime risk, particularly by applying modern research designs and methods such as advanced cognitive computing using machine learning (ML) [5–13].

Cybercrime can be catastrophic. It takes only one malware virus on a company website to result in significant financial losses through extortion demands. For example, the Russian-backed Darkside cybercriminals cyberattacked the Texas USA-based Colonial Pipelines (Colonial) on 7 May 2021 [7]. Colonial transports gas, jet fuel, and other petroleum products through pipelines (as compared to slower trucking, waterway shipping, or railway logistical methods). When Colonial was shut down, the consumer transportation system was impeded due to the higher fuel costs. Colonial paid a $4,400,000 USD Bitcoin ransom to fix the problem, and their corporate image suffered.

### 1.1. Research Rationale Driving the Current Study

Cybercrime is costly to organizations. Raimi [2] found businesses worldwide suffered significant financial losses due to cybercrimes. Parikh [3] claimed $6 trillion USD was spent globally to mitigate cybercrime. The U.S. Federal Bureau of Investigation (FBI) [4] projected that the total cost of cybercrime will be $10.5 trillion USD by 2025.

Trim and Lee [1] asserted that more interdisciplinary cybersecurity research is needed. Martineau, Spiridon, and Aiken [5] revealed from their meta-analysis of 8020 papers, that cybercriminal research is lacking, most of the methods applied were too basic or not explained well enough, and practitioners have not been collaborating with scholars. Almansoori, Al-Emran, and Shaalan [10] published a systematic review of 39 studies from a collection of 2210 relevant papers published from 2012–2021. They argued that self-report surveys for data collection were not reliable. They recommended that researchers collect sources of data beyond self-report surveys and apply ML instead of traditional statistical methods. Almansoori et al. [10] concluded that 56% of cybercrime studies were at the organizational level, so they recommended that researchers ought to study individuals who cause cybercrime rather than analyze the victims or the organizational level factors.

### 1.2. Literature Review of Empirical Studies Focused on Cybercrime Risk

Hiremath, Shetty, Prakash, Sahoo, Patro, Rajesh, and Pławiak [6] developed an innovative approach to identify cybercrime vulnerabilities in customer-facing websites and applications. They applied ML Python coding and Microsoft's Power BI to analyze uniform resource locators (URLs) embedded within an organization's website. It was an action research single case study to analyze an organization's website. They identified cybercrime risk by examining URLs on the case study organization's website. They produced impressive effect sizes, high quality indexes, and a comprehensive feature correlation matrix from the ML. Their technique for identifying cybercrime risk by examining URLs on websites can be leveraged in the current study.

Dalal, Lilhore, and Faujdar [9] were the second-most relevant empirical study of cybercrime risk in the literature. They developed two supervised ML models, the first using Chi-square automatic interaction detection (CHAID) learning tree analysis and the second using the Support Vector Machines (SVM) technique, to predict cyberattacks on organizational Internet of Things (IoT) assets. They created a large dataset (N = 123,404 records) to simulate two cyberattack scenarios (see Figure 1, p. 5, and Figure 5, p. 11 in [9]). The first scenario was a website copier IoT being cyberattacked with 470 simulated cybercriminal attempts. The second case was a simulated HTTP Slowloris Distributed Denial of Service (DDoS) cyberattack on three servers until they were compromised. They claimed the ML CHAID training model was the best. It contained 20 features, 6 levels, and 77 nodes. It has a "90.17% accuracy level overall" (p. 11) for the first scenario and "99.72%" (p. 12) accuracy for the second. The individual quality estimates were broken down by type of IoT attack in the data, which ranged from 25–100%. Their ML ideas could be applied to human behavior using big data instead of simulated data.

Schoenmakers, Greene, Stutterheim, Lin, and Palmerbles [11] took a different approach in their empirical study. They investigated the cybersecurity-related behavior of 21 information technology (IT) security employees. They used a pragmatic ideology, an inductive strategy, grounded theory, and a thematic analysis research method. They collected a small sample of data using structured interviews. They found the thematic analysis results could be explained using the Protection of Motivation Theory (PMT) and the Extended Parallel Process Model (EPPM). This study was relevant because Schoenmakers et al. [11] presented a novel idea to use motivation theories from the psychology discipline to assess cybercrime, which can be leveraged in the current study. Furthermore, their idea could be extended in two ways. Unsupervised ML techniques could be used to analyze big data to overcome the inferential limitations of the small sample size faced by Schoenmakers et al. [11]. ML natural language processing and classification techniques could uncover similar insights as compared to labor-intensive thematic analysis. However, ML can analyze petabytes

of big data instead of being constrained to a small sample, and the ML processes can be completed in minutes, not months.

Kranenbarg, van-Gelder, Barends, and de-Vries [12] used the MANOVA parametric statistical technique. They attempted to get into the mind of the cybercriminal by comparing the personality and demographic factors of prisoners to develop a cybercrime personality profile. They obtained a large sample size of 512 participants, consisting of 261 online and 260 offline prisoners charged with cybercrimes. However, later, they stated their sample size was 1033 (62% were male; the average age was 44, with a standard deviation of 15). They obtained the sample through cooperation with the public prosecutor's office in the Netherlands and paid the participants a token amount for their responses. They leveraged the HEXACO personality construct. HEXACO means honesty-humility, emotional, extraversion, agreeableness, conscientiousness, and openness to new experiences. In the HEXACO model, the emotional factor is equivalent to neuroticism in other personality theories. Although the personality factors were statistically insignificant, the demographic characteristics of male gender, low education (such as grade school), and being a second-generation immigrant were significant. Their idea of testing personality theory to detect cybercrime was novel—it can be applied to the current study where a larger sample size is available.

Another relevant empirical study of cybersecurity was published by Cram and D'Arcy [13]. They applied structural equation modeling (SEM) to data collected from three waves of surveys. They constructed their own model using cybersecurity and organizational legitimacy constructs. They found that negative cybersecurity legitimacy mediated the positive relationship between top management support and cybersecurity policy compliance and between cybersecurity inconvenience and cybersecurity policy compliance. They claimed that employees who judged cybersecurity activities as not legitimate would be less likely to comply with cybersecurity policies. In other words, to prevent cybercrime at the workplace, cybersecurity managers need to convince employees that cybersecurity compliance is important and legitimate, even to overcome inconveniences. Their research was a replication of existing knowledge but still important, as the results affirmed that employees need additional managerial stimulation to comply with cybersecurity policies as company policies, are often perceived as a waste of time.

### 1.3. Literature Review of Non-Empirical Studies Focused on Cybercrime Risk

Martineau, Spiridon, and Aiken [5] published a literature meta-analysis and then developed a proposed future research model that can be leveraged by the current study. They named their concept the cyber behavioral analysis (CBA) model, which applies traditional criminal profiling methods to cybercrimes. They alerted readers to be aware of poor sampling techniques. For example, one study of 3808 students aged 16–19 found that 47% were classified as guilty of cybercriminal behavior. That study should have sampled actual cybercrime victims or potential cybercriminals, not students.

Almansoori et al. [10] reviewed the literature and summarized the common *a priori* theories applied in empirical studies to examine cybercrimes, which can inform the current study. PMT was the most applied construct in 17 of their reviewed studies—an empirical example of PMT was reviewed earlier in the Schoenmakers et al. [11] paper. PMT shows why individuals (usually employees) respond to a supervisor or a bully out of fear, based on threat appraisal and the ability to mitigate or cope with the event. A disadvantage of PMT is that it is individualistic and tacit, so it does not reflect social peer norms, past experiences, or other cognitive variables influencing decision-making in an organizational context. Six of the studies applied Technology Threat Avoidance Theory (TTAT), which tries to rate the employee's awareness of technology threats along with their motivation to avoid the threats, and the construct assesses risk tolerance as well as peer social norm influences. While TTAT was found to be significant in explaining employee behavior toward cybercrime malware, it is technology-driven, and it does not examine the motivational propensities to create or accept a cyber risk. The General Deterrence Theory (GDT) was leveraged in four studies.

GDT, as it sounds, features a negative factor structure. In GDT, negative countermeasures such as penalties or fines are applied to discourage behavior that could lead to an actual perpetuated cybercrime or to cause negligence behavior that would allow a cybercrime to be committed.

The most popular constructs Almansoori et al. [10] found were the Theory of Reasoned Action (TRA), the related Theory of Planned Behavior (TPB), and the Theory of Controlled Behavior (TCB), which includes the Unified Theory of Technology Acceptance (UTAM)/Unified Theory of Acceptance and Usage of Technology (UTAUT). TPB/TCB are considered reliable and valid to measure individual behavior motivations based on attitude and subjective norms from peers. The UTAM/UTAUT and its successors address the technology version of TCB, such as the usefulness, resistance, and ease of use of smartphone applications. However, the above theories require social norm factors in the model or it will not function correctly, and the construct does not specify cybercrime as an outcome event. Instead, the dependent variable was awareness of a specific cybercrime event. Another variation of TCB was the Threat Avoidance Motivation (TAM) model, which was applied in three of the reviewed studies. TAM assesses employee vulnerability, risk severity, and risk avoidance, but it fails to analyze if or why an employee wants to create a cybercrime risk overtly or inadvertently.

The remaining theories identified by Alamansooria et al. [10] in their literature review were applied once, including the five-factor personality theory (FFPT). The most relevant finding of Almansooria et al. [10] was that personality theories are now being used to analyze cybercrime, although traditional statistical techniques were being applied but not ML. A good example of that was the Kranenbarg, van-Gelder, Barends, and de-Vries [12] study, where HEXACO was applied using MANOVA—HEXACO is practically similar to FFPT. Interestingly, Almansooria et al. [10] suggested trying ML. ML is useful for analyzing big data that is too large for traditional statistical software [14,15], and ML as a technique is ideal for exploratory type research designs [16–21].

Lickiewicz [8] developed a cybercriminal psychological profile conceptual model based on the CBA construct by using an abductive literature review approach, which is often accompanied by a pragmatic research ideology. The abductive approach means to find studies that are likely to support a research question (RQ) and be complementary to one another, rather than being the driving force of the RQ as in the deductive positivistic ideology where hypotheses are developed. As a further contrast, an inductive approach means to use the most relevant theories from the literature review to explain the results of the analysis, rather than to use the theories to inform the research design and techniques selected (as in abductive and deductive approaches). The key factors in Lickiewicz's model were: biological (e.g., medical burdens), relationships, intelligence (ability to strategize), personality, social norms, technical ability (ability to program execution), internet addiction, motivation, attack method (social engineering, others), process maturity (organized, no evidence left), and attack efficacy (did it work).

Lickiewicz [8] leveraged the *a priori* FFPT from the psychology discipline to develop the central component of his model. He argued that an elevated level of openness to new experiences and neuroticism should correlate with a strong motivation to hack into a computer system or to defraud a vulnerable victim. Lickiewicz thought that significant neuroticism could also explain why cybercriminals usually prefer anonymity yet brag about criminal accomplishments through avatars and terrorist groups. He suggested that employees in an organization may have a neurotic tendency to despise supervisors, create work conflicts, and violate cybersecurity policies out of boredom. Additionally, according to Lickiwiecz [8], a cybercriminal remains open to new experiences to learn novel ways to break into organizational security constraints.

*1.4. Personality Theory Overview from the Literature*

Following the ideas of Lickiewicz [8], Alamansooria et al. [10], as well as Kranenbarg, van-Gelder, Barends, and de-Vries [12], FFPT was selected for the current study. FFPT is

also called NEO, OCEAN, Big Five, or 'personality traits' in psychology literature. NEO is an abbreviation of the first three dimensions: neuroticism, extroversion, and openness to new experiences. OCEAN is an abbreviation of the five factors, but in a different order: openness, conscientiousness, extroversion, agreeableness, and neuroticism. The Big Five simply refers to the five factors. Personality theory has been shown to reliably predict 50% of human behavior, including that of employees [8,21,22]. Therefore, FFPT is relevant because it can be used in the current study.

The FFPT construct was developed in the 1980's by numerous psychologists, including the key contributors Goldberg [21] as well as Costa, McCrae, Zonderman, Barbano, Lebowitz, and Larson [22]. Each of the five distinct factors contains 10–20 items to identify low-to-high personality orientations—the items are transformed into questions in surveys. The items have been adapted by researchers and translated into other languages. A 1–5 scale is commonly applied to each item, where 1 is low agreement with a factor item and 5 is high agreement. Some items are reverse-coded to test social desirability (honesty) and must be normalized by rescaling them to the above 1–5 format.

The FFPT item scores are typically averaged to form a factor score. Researchers have typically applied factor analysis, or SEM, to survey data collected from a FFPT instrument to reveal semantic associations between the items within each factor, which in turn can be used to describe a person's personality profile. Companies sometimes require job applicants to take FFPT surveys to facilitate selecting what they view as good candidates, such as having low neuroticism but high agreeableness and conscientiousness [21,22]. This implies ML could analyze big data text using the FFPT.

Neuroticism in FFPT is an emotional balance continuum. Strong negative behaviors such as aggressiveness, deception, anxiety, or depression are considered emotional stability—high neuroticism. On the other end of the scale, attributes such as calm, relaxed, and optimistic are considered emotionally stable traits—low neuroticism. Example items are 'I get stressed out easily', 'I often feel blue', while reverse-coded items are 'I am relaxed most of the time' and 'I seldom feel blue' [21]. Adverbs such as 'blue' were from earlier generations, so modern terms such as 'depressed' will be substituted in the current study.

Agreeableness can be thought of as having a good relationship with others or accepting directions. A person with an elevated level of agreeableness would be willing to compromise their interests to preserve harmony. At the opposite end of the continuum, a person with low agreeableness would be classified as disagreeable, unconcerned with following orders or directions. Studies have found that prominent levels of agreeableness are positively correlated with high quality relationships in a team, and it predicts transformational leadership skills [21,22]. Example items are 'I am interested in people', 'I take time out for others', a reverse-coded item is 'I am not interested in other employee problems' (adapted from [21]).

The openness to new experience FFPT feature, or openness, refers to a willingness to try something new. High scores of openness have been associated with creativity, imagination, curiosity, and willingness to try something different. Some researchers point out that people with high openness ratings may be perceived as more likely to engage in risky behavior. The low level of this attribute means being conservative, unwilling, or slow to try new things [21,22]. Example items include 'I have a vivid imagination' and 'I am full of ideas', while a reverse-coded item is 'I am not interested in new ideas' (adapted from [21]).

The conscientiousness factor refers to acting dutifully. Low conscientiousness implies being unreliable or sloppy, while a high coefficient suggests being pedantic, being detailed in work, or a lack of flexibility. An example of a conscientiousness item is 'I pay attention to details' while a reverse-coded item is 'I make a mess' (adapted from [21]). Prominent levels of conscientiousness were associated with accuracy at work [22].

Extraversion describes being social or not, according to the FFPT construct. A high reading of extraversion is associated with someone who enjoys interacting with people, and they like to talk. The low end of extraversion is equivalent to being an introvert.

Introverts tend to not engage in social activities, and they may be perceived as being shy or depressed. Some researchers have shown a positive impact on introverts—that they are more self-regulated and will act without being told. Example items are 'I feel comfortable around people', 'I start conversations' while typical reverse-coded items are 'I do not talk a lot', 'I do not like to draw attention to myself'.

### 1.5. Literature Summary and Research Question in the Current Study

As a summary of the above literature review, there were several extant papers focused on cybercrime risk or other relevant topics, such as how to apply ML to big data. Five relevant papers informed the design of the current study. Almansooria et al. [10] reported that the FFPT construct had been utilized outside of the psychology discipline to examine cybercrime risk. Lickiewicz [8] provided a conceptual model to identify potential cybercriminal behavior using the FFPT construct. Kranenbarg, van-Gelder, Barends, and de-Vries [12] demonstrated how to apply HEAXCO (similar to FFPT) to detect cybercrime, but they used a survey to collect data and MANOVA, not ML.

Two other empirical studies provided additional value to the current study by demonstrating how to apply CHAID in ML to analyze big data and how to interpret URLs on a website to predict cybercrime risk. Dalal, Lilhore, and Faujdar [9] provided essential guidelines for how to apply the CHAID technique in ML on big data. Hiremath et al. [6] contributed the essential technique for using ML to detect cybercrime risk based on parsing URLs on a company website (which they implied was big data).

However, there was a gap in the current literature. There were no studies at the time of writing examining how personality theories could predict cybercrime risk by applying ML to analyze retrospective employee behavior big data from a company intranet. Subsequently, the aim of the current study was to apply ML to a retrospective big data source of employee behavior, ethically obtained from a case study organization, to identify cybercrime risk by leveraging the FFPT (personality theory). The research question was: Could FFPT be integrated into ML to identify organizational cybercrime risk by analyzing retrospective employee behavior big data?

## 2. Materials and Methods

### 2.1. Research Design Summary

The research design is conceptually summarized in Figure 1. The choices selected by the author were highlighted in red and bolded. Steps 5–7 encompass the ML phases, which were compressed from the traditional four-phase ML method of plan, build, evaluate, and implement. The research design steps in Figure 1 are enumerated below:

(1)     Design study (ideology: pragmatic, strategy: abductive, nature: exploratory);
(2)     Review RQ-related literature (abductive strategy focused on relevant topics);
(3)     Identify the population (sample technique, unit of analysis, analysis level, data);
(4)     Methods and materials (data types, relevant method ML, ethics, permission);
(5)     ML: methods plan phase (semi-supervised ML, some labels, classification);
(6)     ML: explore data phase (data collection/cleaning, CHAID decision tree);
(7)     ML: develop models phase (credibility, validity: 20-folds, reliability checks);
(8)     Document research study (quality assurance, dissemination to journal).

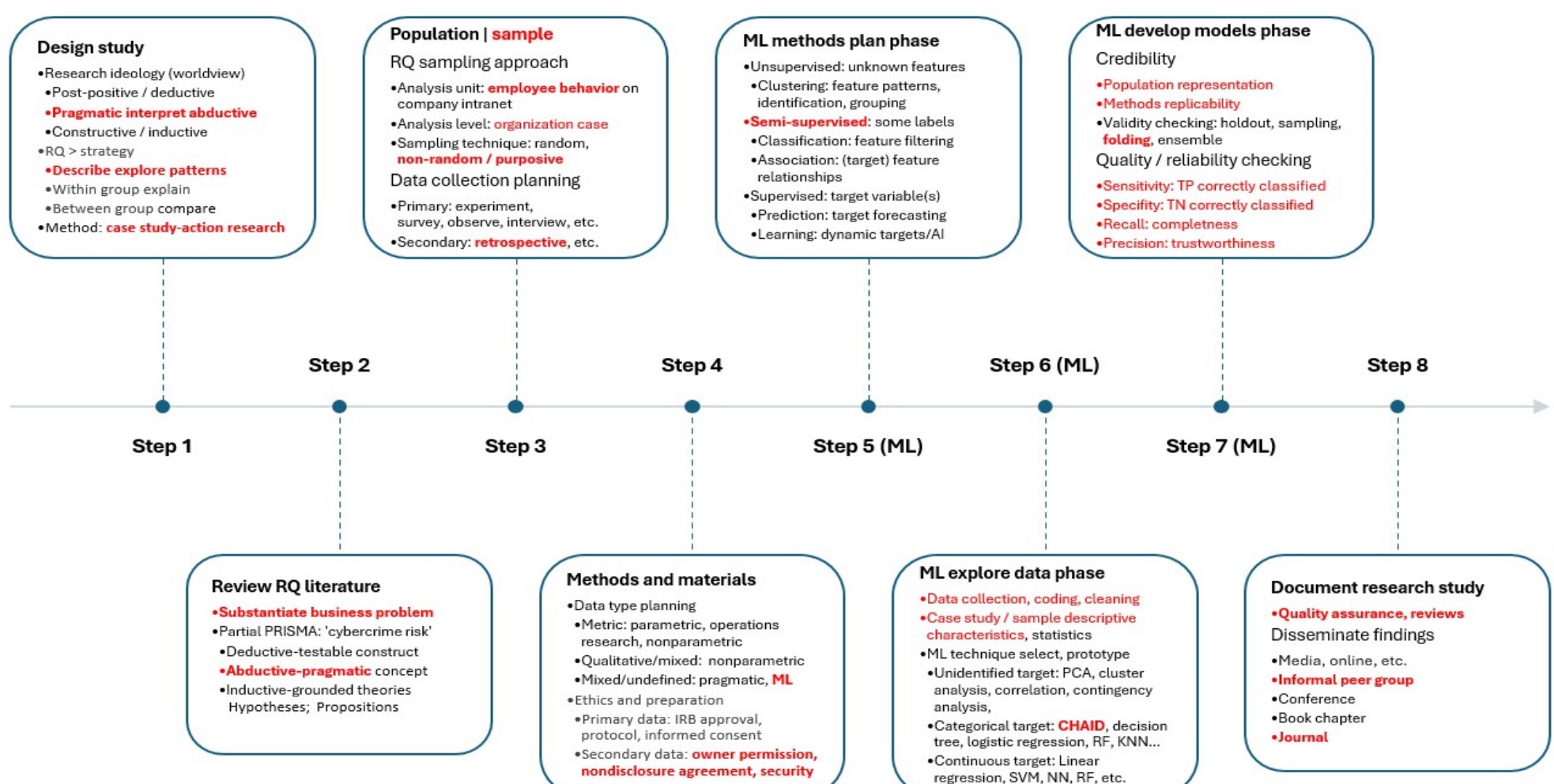

**Figure 1.** Research design developed by the author and applied to the current study.

### 2.2. Research Ideology, Strategy, and Overall Methodology

Step 1 of the methodology was to externalize the author's ideology or worldview being applied for the study, which was pragmatism. A researcher's ideology is described by Lim and Chai [16] as existing on a mental continuum from positivism on one end, towards pragmatism in the middle, and constructivism at the other end. The unit of analysis was identified as employee behavior through typing on the company's SharePoint intranet. The level of analysis was the case study organization. The RQ was focused on integrating the FFPT personality theory construct into ML to analyze employee behavior big data to explore how cybercrime risk might be identified.

The overall methodology was a single case study embedded within an action research context, based on [16]. Action research (AR) is different from formal scientific experiments and traditional scholarly methods such as surveying or multiple case studies because in AR, the researcher is actively engaged in the business project while in parallel managing the scholarly research [16–18]. Here, the author conducted all the research and assisted the organization, including the ML/R/Python programming. The single case method is well known in scholarly literature, and it has sometimes been integrated with AR business projects [16,17]. Organizations and academic scholars have recognized the mutual benefits of AR, where *a priori* theories are conjoined with workplace behavioral projects to solve important business problems [17,18].

Big data is often misunderstood, and each discipline has a unique definition. In the business discipline, big data can be defined in a simplistic way as the five V's [14]. Big data five vs. refer to substantial value (importance to decision-makers), strong velocity (rate of added information arriving or being created), clear veracity (truthfulness), high volume (amount in storage), and complex variety such as mixed data types [14]. Hiremath et al. [6] was the only relevant study identified in the literature where the AR case study method was applied to analyze big data (a company's website).

### 2.3. Literature Review Approach

In step 2 from Figure 1, the literature review was focused on relevant empirical studies first and relevant literature reviews second. An abductive strategy was applied by selecting studies with useful ideas rather than creating a complete context for hypothesis testing, such as the deductive perspective of positivist designs. Inductive strategies were not relevant for the current study because the FFPT had been identified.

To accomplish that, ProQuest, World of Science, and Google Scholar literature databases were searched using variations of "cybercrime risk", "behavior big data", and "cyber security" keywords, and later by adding "ML", "machine learning", "big data", "personality theory", "five factor", "NEO", and "OCEAN" to the search terms.

### 2.4. Population and Sampling

In step 3 of the research design in Figure 1, the intended population of interest was identified as organizations (public or private) involved in the financial information technology software programming industry. The sample was a particular case study organization. The sampling strategy was purposeful because it was inexpensive and because it was one of the few authorized big data sources available to the author. Many organizations were reluctant to allow research scientists to study their internal data, even for mutually beneficial reasons. The sampling strategy was nonrandom theoretical because a competitor in the same industry as the organization had recently suffered a ransomware cyberattack, so cybercrime risk was considered a valid phenomenon in the underlying population, being the case study company's industry.

The case study company was a large multinational corporation in the financial technology software application development business. Their revenues were over $1 billion USD, with offices in the USA, India, Australia, and South Korea. They had approximately 12,321 employees. Since the big data was produced prior to the study and was not created specifically to answer the RQ for the current study, it is known as secondary retrospective

data. The author argues that collecting primary data would have been impossible due to the anticipated risk of participant self-report bias and too costly due to the size of the big data. Alternatively, the author asserts that secondary retrospective big data contains honest, reliable, and better indicators of employee behavior.

### 2.5. Ethics and Big Data Access Permission

In step 4 of the research design (Figure 1), the expected data types governed which family of methods and techniques would be selected to answer the RQ. ML was chosen as the best choice for an overall technique. This decision was made because the ideology was pragmatic, the *a priori* FFPT materials from the literature were being used in an abductive strategy, and big data was the source with initially unlabeled features. Since the author intended to create some features from the data by using the FFPT and URLs, the family of semi-supervised ML techniques were identified as appropriate.

In terms of research ethics, the case study company held a high trust relationship with the author, and the organization was willing to provide some of their data to the author to advance scholarly science. The author signed a non-disclosure agreement with the case study company, which prevents discussing or releasing anything concerning the project except what the organization authorizes. Ethical clearance and permission to use the big data were given to the author by the case study company, and a redacted copy of the authorization document was given by request to the MDPI BDCC Assistant Editor.

### 2.6. ML Technique Planning

In research design step 5 from Figure 1, the ML methods plan phase consisted of experimenting with prototypes of the anticipated big data to select the correct technique [19,20]. Experimenting with prototypes in ML is different as compared to an experimental research design. The former is common in ML (where the researcher generally has a pragmatic ideology) to fine-tune technique selection. An experimental research design is a completely different research design, where the researcher holds a positivistic ideology, a deductive strategy is applied to the literature review, experimental and control variables are identified, hypotheses are developed, human participants are recruited (after approval), informed consent is obtained, the experiment is conducted in a controlled context, and parametric statistical techniques are applied.

The author planned to integrate the FFPT construct into ML to create continuous data type factor labels from the employee behavior big data. The author also planned to create a cybercrime risk binomial target variable from the big data. The most appropriate family of ML techniques for the current study were the semi-supervised classification ML routines that are specialized for metric labels and binomial target variables. The best ML technique was CHAID learning tree analysis, as demonstrated by [9].

### 2.7. Big Data Exploration with Selected ML Techniques

In research design step 6 from Figure 1, the ML explore data phase was where the training model was developed. First there was the data collection process, which involved coding and cleaning the data. The big data was processed to remove confidential company information or employee identifying attributes and then preprocessed using ML Python to develop the features and target variables.

Microsoft's Office software and Microsoft's Azure platform were in use by the case study company. The author also utilized Microsoft's Azure AI, ML, and cognitive computing services. To help others, the author can share that Microsoft [15] recently opened their AI and ML cognitive services to the public, providing 25 ML tools and at least 55 services free [see: https://azure.microsoft.com/en-us/pricing/free-services/ (accessed on 1 March 2024)]. Microsoft stated the AI/ML programming libraries contained emotion/sentiment detection, vision/speech recognition, and language understanding utilized by their Bing, Cortana, and Skype Translation products [15]. Google has comparable services.

The research programming environment consisted of the Microsoft Azure CLI ML extension version 2 and the Python SDK Azure-AI-ML version 2. Specifically, these libraries were accessed: azure.ai.ml, azure.identity, azure.ai.ml.entities, azure.ai.ml.constants, and other general-purpose routines commonly leveraged for research programming tasks. The author primarily used Jupyter Notebook as a structured programming editor to prototype and then train an ML model using Pandas Conda commands. This environment was considered comparable to what Dalal et al. [9] used: Python 3.6, NumPy, Pandas, Scikit-learn libraries, and more powerful than a 1 GHz CPU with 2 GB of RAM. The author initially set up a workspace instance attached to a serverless computer to offload the lifecycle management to Azure Machine Learning. Then, a datastore was defined in the cloud as an Azure Data Lake, which was where the case study company securely placed the extracted data. The same work environment and resources were reused by the author for subsequent iterations of the model building. The AI/ML documentation is stored at: https://learn.microsoft.com/en-us/azure/machine-learning/ (accessed on 1 March 2024).

The case study company created an application programming interface (API) with the help of the author to clean up the big data. The API extracted text blobs from the big data source, first ensuring confidential protected information was removed by using lookup registries of employee personal identifier constructs (usernames, identification numbers, etc.) and trade secret phrases for a year. The API returned only text, no images, graphics, or attachments, from employee postings or Microsoft Office applications. The data lake held a trillion datasets and a single file up to a petabyte in size. The size of the current study's cleaned big data was estimated to be 8.8 gigabytes.

Once the API was operational, the next task was to integrate the FFPT construct into ML. First, the FFPT items were adapted from the work of Goldberg [21] and Costa et al. [22]. Table 1 contains the item keywords adapted for each FFPT factor. Reverse-coded items were not used because they were intended to check social desirability—in the current study, there were no survey self-reported data, only retrospective big data. The FFPT items were loaded into an ML Python lexicon array, structured by factor.

**Table 1.** Adapted FFPT Personality Factor Items Integrated into the ML Lexicon Array.

| Neuroticism | Openness | Agreeableness | Conscientiousness | Extroversion |
|---|---|---|---|---|
| stressed | create creative | interested | prepared | party parties |
| worried | imagination | sympathize | detail detailed | conversation |
| upset | ideas | comfort | exact | attention |
| pissed | complexity | gratitude | accurate | talk discuss |
| moody | new newness | follow | plan | friends |
| angry | try | accept | attention | social |
| revenge | experiment | listen | quality | fun |

The Vader English library was selected for the tokenization because it is rule-based and effective in previous ML studies [15,18,19]. The WordNet Lemmatizer in the Python library was leveraged for further normalization because it has a large pre-trained dictionary of English words that can provide synonyms. The result was that unimportant connecting words, called stop words, were removed (e.g., 'but', 'and', 'in', 'or', etc.), and modern American slang synonyms were added, such as 'pissed' to represent 'outraged'. To facilitate further processing, all words were forced to be lowercase. Thus, the normalization condensed each text blob into a succinct group of words, in lowercase, without punctuation, in hopes of capturing the essence. The sentiment analysis was performed using natural language processing by comparing the normalized big data text blob records to each FFPT item per factor.

FFPT coefficient scores were calculated on a 0 to 1 continuous scale per item. The item scores were averaged to obtain a factor coefficient score. For example, the first item in the FFPT neuroticism factor from Table 1, 'stressed out', was compared to the big data record, to generate the item level score. If the big data record contained the tokenized keyword

'stress' in the past, present, present-perfect, or future verb form, this would be scored 1. If the big data contained the noun form with adjacent keywords (e.g., 'stressed, wow am I ever…') it would be scored 1. Averaging the item coefficient scores per factor resulted in all coefficients ranging from 0.1 to 0.95 for every FFPT factor.

*2.8. Big Data Analysis, Training Model Development, and Evaluation with ML*

Step 7 in the research design of Figure 1 was 'ML develop models'. The first task was to develop a training model from the normalized and tokenized big data. After that, the quality, validity, and reliability of the model were measured.

Credibility in step 7 was partly achieved through how the research was designed and partly justified by excellent quality scores based on established benchmarks. Credibility was achieved in the current study by fully describing the methods so that another researcher could recreate or replicate the current study and understand how it was accomplished scientifically. Additionally, another concern for credibility is how well the population was identified and how well the sample represented the targeted population. In the current study, the targeted population was made clear, and the sample was taken from the industry. This allows generalizations based on the findings of the current study to be made with confidence by readers and other researchers.

In step 7, ML validity checking was applied to the model. In ML, validity checking is performed by separating the data used to test the model from the data used to train the model. The best practice is to divide the data into three groups using the 80:10:10 rule: the first 80% for training model data, the next 10% for model testing validity/reliability, and the final 10% to validate the final model after tuning in the future [19,20]. It is acceptable to use a 90:10 rule in exploratory studies, with 90% of the data allocated for building a training model and the remaining 10% allocated for validity/reliability testing of the current model as well as for future fine-tuned models [20]. The key issue is how the test dataset is created from the data because random pulls may not be representative of the big data due to the limitations of statistical probability sampling with replacement [19].

There are four common approaches in ML for allocating a test dataset to perform validity/reliability checking on a training model: sampling, holdout, folding, and ensemble. The first three involve allocating a proportion of the entire data using the 80:10:10 or 90:10 rule discussed above. The ensemble approach involves using multiple ML methods to create multiple models, combined with one of the validity/reliability checking approaches (e.g., sampling, holdout, or folding) applied to every model with the results averaged or the overall best result selected [20]. Another approach not mentioned above because it is not practical for big data is the hold-out or leave-one-out, where iterative testing is performed using all but the withheld records, selecting new records each time, which is reliable because it eliminates errors by chance, but it will be extremely time consuming even with powerful computers [19]. The sampling approach is the least robust because it takes random records to create the test (and optionally the validation) datasets, with or without replacement, but there is no assurance the test record distribution values will identically match the distribution values of the training data unless stratified selection takes place, and even then validity/reliability can suffer from pure chance [20]. However, sampling is convenient and may be practical for small datasets. In simple terms, folding is also known as cross-validation, a variation of the hold-out mentioned above, where 10% or another proportion of the records are held-out from the training model data, and this process continues for 10 to 20 times to test the validity/reliability [19,20]. This is considered more robust than sampling and less time/resource intensive than the ensemble approach [20]. A decisive point to mention is that many ML techniques generate quality score coefficients during building the training model, but these are intended to be rough progress indicators—the quality measures must be taken after the training model is finalized and by using the selected approach.

The folding approach was selected for the current study because it was considered a robust accuracy validation/reliability technique by experts [19,20], and it works well

for models with dichotomous target variables such as cybersecurity risk. A combined folding and stratified sampling approach was used, with a 20-fold cross-validation of the training with the test data and reporting the average over all classes as the final accuracy coefficient score. The author asserts that this approach is robust because it surpasses the 10-fold cross-validation resampling technique recommended by Lantz [19].

Step 7 of the ML training model validity/reliability checking involved calculating several score coefficients and then evaluating the scores against *a priori* benchmarks from the literature review or by expert ML practitioners [19,20]. Here, we can call these quality measures in general or a coefficient if referring to a specific score. This task started by calculating a confusion matrix from the ML training model results. Some coefficients are conditional probabilities, and others are formed through regression residual calculations. The confusion matrix is a contingency matrix or table listing the frequency counts or proportions of the true versus false comparisons using the training model algorithms on the test data (e.g., on the folding test records). The terms true and false here do not mean cybercrime risk yes or no, but instead whether a predicted value was correct (true) or not (false). Some ML practitioners suggest displaying a confusion matrix to increase credibility so that other researchers may clearly see the values underlying the coefficient scoring, which was performed in the current study (see discussion section below).

The first ML quality measure calculated from the training model confusion matrix was the classification accuracy (CA). CA is the proportion of correctly classified records, correct positives, and correct negatives. This is found as the cumulative percentages of the diagonal line on the confusion matrix, from top left to bottom right. The mean absolute error, or MAPE, was approximated by using the formula 1-CA.

The sensitivity quality measure shows the true positives, which were correctly classified. Sensitivity calculates the proportion of the positive predictions using the training model algorithm on the folded test data in the current study, which were correctly classified [19]. The reader could think of this as the proportion of true positives correctly identified from the test cases divided by the total number of positives in the folded test data (records correctly classified as true positives and records incorrectly as classified positives that were false, known as false negatives). In formula format, it would be: True positive records/(True positive records/False positive cases). Sensitivity must be balanced as a tradeoff with its reciprocal specificity, discussed next.

The specificity quality measure encapsulates the total negatives correctly classified, the true negative rate [19]. The reader can think of specificity as the proportion of negative records that were correctly classified by the training model out of all negative records in the test data set. The formula is approximated as: true negatives/all negatives in test data. As noted above, specificity is balanced with sensitivity, changing one impacts the other.

The recall ML quality measure indicates how complete the training model test results are, which reflects the number of true positives over the total number of positives [19]—this is the same formula as sensitivity, and the same interpretation applies! The precision ML quality score is an indication of training model trustworthiness. The precision coefficient is the positive predictive value [20]. It is calculated as the proportion of positive records from the test data using the training model algorithm that were correctly identified as truly positive. This is an important coefficient for business decision-makers because it shows how often a ML training model is correct or trustworthy, because untrustworthy models could result in many lost customers over time [19]. The pseudo formula for ML precision is: true records/(true + false records).

The F-measure is an overall score for evaluating all the above quality measures in one index. F-measure evaluates the ML training performance against the test data by combining the precision and recall coefficients into a single index by calculating the harmonic mean. The harmonic mean is used rather than the more common arithmetic mean since both precision and recall are expressed as proportions between zero and one [20]. The pseudo formula for F-measure is: $(2 \times \text{precision} \times \text{recall})/(\text{recall} + \text{precision})$.

The next ML training model quality measure calculated in step 7 was the area under the receiver-operating curve (AROC), sometimes abbreviated as AUC. AROC is a scatter plot of the false positive rate (or 1-specifity) on the *x*-axis versus the true positive rate or sensitivity on the *y*-axis, plotted as an area against a superimposed cutoff slope line from 0.0 to 1.1 (zero quality). According to Lantz [19], the AROC curve should be used to evaluate the quality tradeoff between the detection of true positives while avoiding false positives—in other words, it should be above the zero quality slope line and the perfect maximum of an imaginary line from 0.0, to 0.1, and then 1.1 (a T-shaped line at the top of the chart). To interpret the AROC, the closer the line is to the perfect T line, the better the training model is in terms of quality by identifying true positives. The coefficient score for AROC is measured as a two-dimensional square representing the total area between the zero-quality line and the plotted line. The AROC benchmarks, according to and adapted from ML practitioner Lantz [19], are:

- 0.9–1.0 = outstanding quality;
- 0.8–0.9 = excellent/superior quality;
- 0.7–0.8 = acceptable/fair quality;
- 0.6–0.7 = poor quality;
- 0.5–0.6 = zero quality.

ML subject matter experts Ramasubramanian and Singh [20] as well as Lantz [19] argued that Jacob Cohen's inter-rater agreement Kappa coefficient statistic is an important error metric to consider for evaluating training models developed to solve business problems because the formula adjusts for chance. For example, in statistical theory, it can be proven that an ML model for a university exam with 80% yes values (positives) and 20% no values (negatives) can be faked with a yes (positive) answer to all questions [19]. The Kappa or equivalent formula adjusts for the expected probability of inter-rater agreement due to pure chance.

In the current study, the Log Loss estimate was calculated. Like the Kappa coefficient, the Log Loss shows how well the training model classified the values in the folded test data as compared to a null model, where prediction would be by chance [20]. Therefore, a lower Log Loss coefficient is desired. A quality score could be generated by taking the reciprocal of the Log Loss index. The following Log Loss benchmarks were adapted from Lantz's [19] Kappa interpretation guidelines for use in the current study:

- 0.00–0.20 = low agreement with the null model, benchmark acceptable score;
- 0.20–0.40 = weak agreement with the null model, baseline acceptance score;
- 0.40–0.60 = moderate agreement with the null model, borderline to poor score;
- 0.60–0.80 = good agreement with the null model, poor score;
- 0.80–1.00 = high agreement with the null model, unacceptable score.

Once the ML model validity/reliability checking in step 7 was completed and the quality scores were considered acceptable, the results were visualized and interpreted. Consequently, the final task in step 7 was to create a word diagram, a node diagram, a scatter plot, and a heat map to visually interpret the results.

The word diagram was created by selecting 50 keywords from the big data associated with cybercrime risk. The learning tree analysis node diagram was created to explain how the big data were classified by the FFPT factors. The scatter plot was developed to contrast the two most important FFPT factors using three parameters. The most important FFPT factors were aligned to the axes, namely, neuroticism on the *x*-axis and openness on the *y*-axis. The color was scaled to match the coefficient score of the cybercrime risk association (lighter yellow shades represented minimal risk coefficients, and darker shades of blue were higher risk coefficients). The symbol was arranged to differentiate data values between no cyber risk—shown as a '0', and a potential cyber risk—shown as an 'x'. A regression slope trend line was superimposed on the plot.

A cluster analysis heatmap was used to contrast and highlight the most frequent instances of the FFPT features related to cybercrime risk, particularly neuroticism and

openness. This can simplify and condense scatter plot data into a smaller diagram by using cluster analysis to group similar data values together (whereas in the scatter plot, each sampled data point is shown). The heatmap was created using the coefficients from the learning tree analysis, but in a unique way, by clustering or grouping incoming big data features and the target variable. A dendrogram was created within the heatmap to show the feature cluster relationships in the context of high versus low cybercrime risk.

### 2.9. Plan to Finalize and Disseminate Research Study Findings

The last step 8 of the research design from Figure 1 was to document the research study. The author followed the American Psychology Association writing style and the recommendations of AR/case study experts [16–19] for conducting the research study. The MDPI BDCC manuscript template and author guidelines were used for the paper. Informal and formal peer reviews took place to support the journal's publication goal.

## 3. Results and Discussion

This section describes and interprets the results from step 6 ML explore data phase (data collection/cleaning, CHAID decision tree) as well as the quality scores from step 7 ML develop models' phase (credibility, validity: 20-folds, reliability checks). First, the quality scores are reported and interpreted using the benchmarks cited in step 7.

### 3.1. Quality Scores from ML Training Model Development and Testing

The first task in checking ML training model quality was to create the confusion matrix, which is shown in Table 2. The first preliminary indication of excellent quality can be observed from Table 2 by looking at the diagonal joint probability values from the top left down to the bottom right (highlighted with bold and red). The no cybercrime risk condition had a 64% for actual and predicted values correctly classified, while the yes cybercrime risk was a lower 24% for actual and predicted values correctly calculated by the training model using the 20-fold test data of employee behavior. The quality scores can be calculated from the confusion matrix using ML algorithms.

**Table 2.** Confusion matrix for cybersecurity risk learning tree training model.

| | **Predicted** | | |
|---|---|---|---|
| **Actual** | **0 (Risk = No)** | **1 (Risk = Yes)** | **Total** |
| 0 (risk = no) | 63.6% | 6.1% | 69.7% |
| 1 (risk = yes) | 6.1% | 24.2% | 30.3% |
| Total | 69.7% | 30.3% | 100% |

The cybercrime risk learning tree training model classification accuracy (CA) coefficient was 0.879, which can be interpreted as excellent quality based on the benchmarks from Lantz [19]. The quasi-MAPE calculated from this would be 12.1%, which is a low error rate given that the population is the financial software development industry, and the AR business problem was to detect cybercrime risks from employee behavior on the workplace intranet. A 12.1% error rate could mean that if the model mistakenly identified an employee behavior that was not a cybercrime risk, this would be discovered without too many repercussions by simply asking the employee to share the details with the security officer so that this could be verified using other data sources (the employee and the security officer doing some research). On the other hand, if the cybercrime risk were missed, it is likely that the organization would have other safeguards in place to catch these missed risks, such as firewalls.

Since the sensitivity rate is the same formula as recall, we can discuss the results together. The recall rate was 0.879, which is considered excellent quality based on benchmarks (see [19]). This 88% recall/sensitivity rate can be understood as the validity of the

training model test results to detect cybercrime risk in the big data of employee behavior using the FFPT construct. We can also see that 12% of the employee behavior cases were not correctly detected, which carries the same implications as discussed above for the quasi-MAPE of 12.1%—in this context, the model would be helpful.

The precision rate from the learning tree training model was 0.879, which is considered excellent quality based on benchmarks (see [19]). This precision result of 88% could be interpreted as the cybercrime risk learning tree model being trustworthy in that it will reliably classify 88% of employee behaviors as not risky, although at the expense of not detecting 12% of the behaviors that could become a cybercrime risk to the organization.

The specificity rate for the training model was 0.834, which again is excellent quality based on industry standards (see [19,20]). We can interpret the specificity quality measure of 84% as indicating how well the actual cybercrime risks were identified from the employee behavior in the big data when using the FFPT construct. On the other hand, this implies that 16% of the actual cybercrime risks in employee behavior big data were not correctly classified. It may be better to use this reciprocal of the specificity rate, 16%, rather than the MAPE rate of 12.1%, as an indicator of the disadvantages of this cybercrime risk learning tree model.

The F1 quality measure from the training model test was 0.879, another excellent/good finding as judged by industry standards (see [19]). The F1 of 88% is important because, as explained in the methods, F1 is an overall index for evaluating all the above quality coefficients in one metric. The Log Loss rate from testing the learning tree training model was 3.68%. This is an excellent score because it infers there was a low agreement between the training model and a null model.

Finally, the AROC coefficient was 0.879, which is a good result, meaning 88% of the results were captured by the cybercrime risk training model in the quality tradeoff between the detection of true risks while avoiding the false risks that may have looked real. We can use the graph in Figure 2 to visually examine this tradeoff between sensitivity and specificity. We can observe from Figure 2 that the green solid line (representing sensitivity versus 1-specificity, after the 20-fold testing and validation) is higher than the dashed baseline of 50%, as it approaches 67%, which indicates an acceptable result but a slightly lower than optimal result according to Lantz [19], where 70% was preferred. Nonetheless, we could attribute the small gap between 67% and the AROC benchmark of 70% to the large volume of big data after 20-fold validation, where only a few of the employee behaviors were likely to be associated with cybercrime risk, in the typical workplace. We can accept this model as usable, but in doing so, we point out that more studies are needed to replicate and retest the accuracy of the cybercrime model.

### 3.2. Cybercrime Risk Keyword Association Analysis

Now that the accuracy, reliability, quality, and credibility estimates of the ML learning tree model have been accepted based on the body of knowledge benchmarks, a visual model has been created. Figure 3 is a diagram of the most common keywords found within the employee behavior big data text blobs, adjacent to the FFPT and cybercrime risk factors. Due to the size of the big data and display space limitations here, only the most important 50 words were selected for the diagram.

This model can be interpreted to reveal the most important natural language keyword associations within the employee conversations and postings where there was a cybercrime risk detected. These words show the other side of the big data, that is, the context of what the employees were writing when the FFPT items were associated with cybercrime risk. However, this does not necessarily prove these keywords mean a higher cybercrime risk due to the font size. The font size merely represents the frequency of all words from various employee conversations or postings in the group of text blobs associated with cybercrime risk. A larger sized word means it was found more often in the cybercrime risk group, but not that a keyword itself is a higher cybercrime risk. Other ML techniques will be used next to identify the cybercrime risk factors.

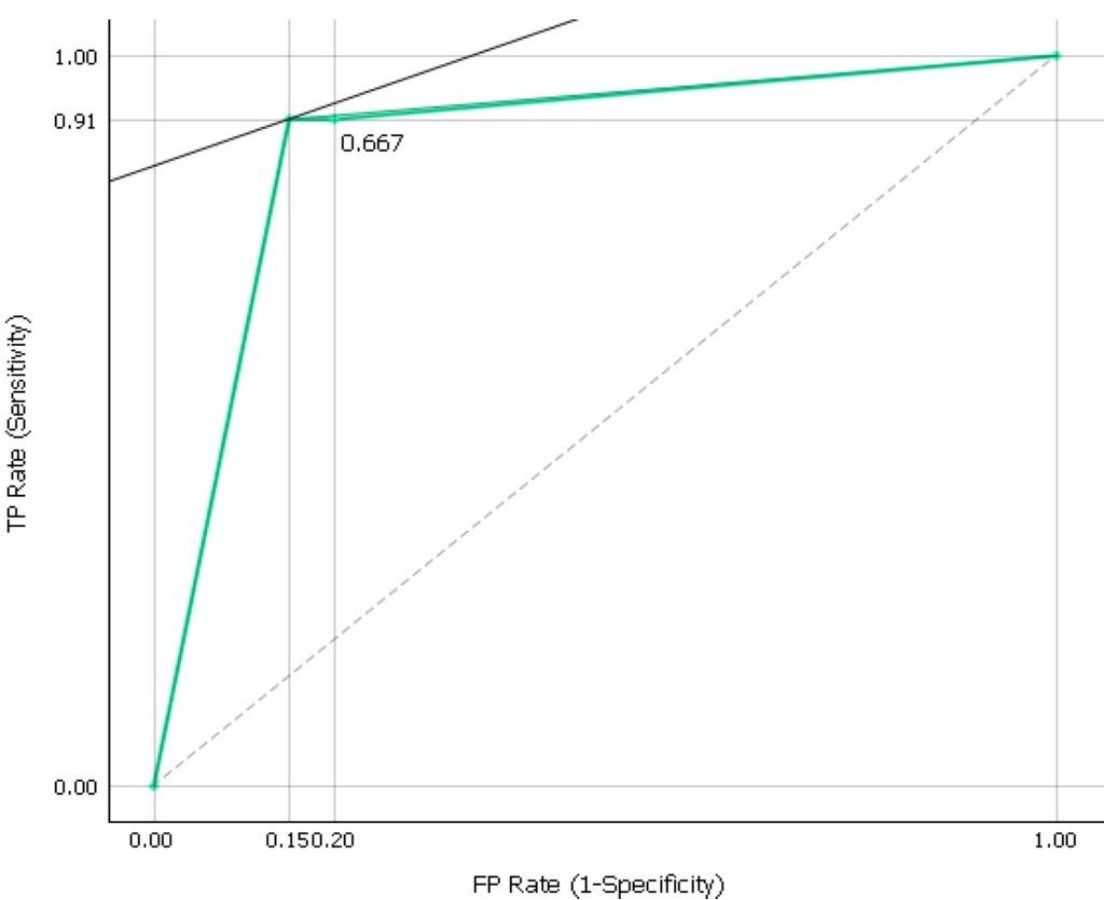

**Figure 2.** Area under receiver operator characteristic in the cybercrime risk training model.

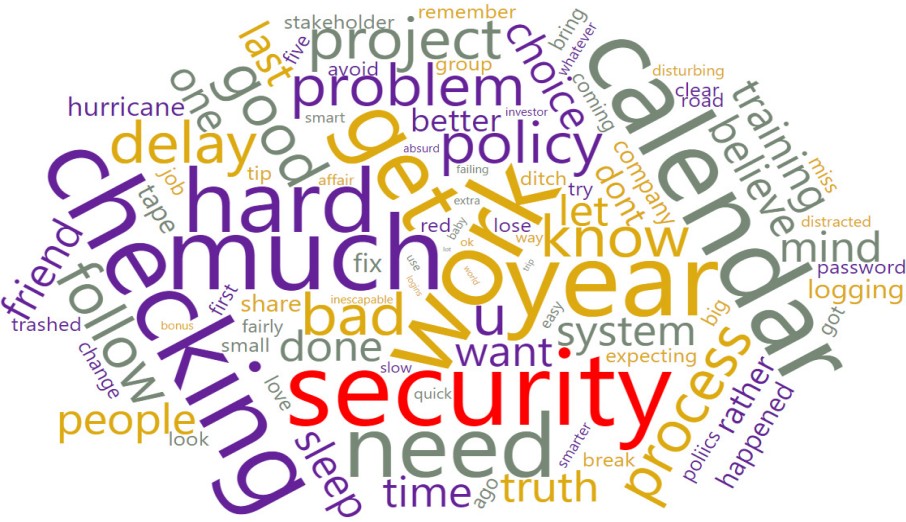

**Figure 3.** Important keywords from employee behavior big data associated with cybercrime risk.

### 3.3. Cybercrime Risk Learning Tree Analysis

The next task was to discuss how the CHAID results could answer the RQ. A decision tree diagram was produced to facilitate this discussion, as depicted in Figure 4. The tree analytics produced 5 branch levels and 8 leaf nodes. Note that there were no thresholds set for the branches or leaves since this was a first-time exploratory study, in a situation where there was scant *a priori* research.

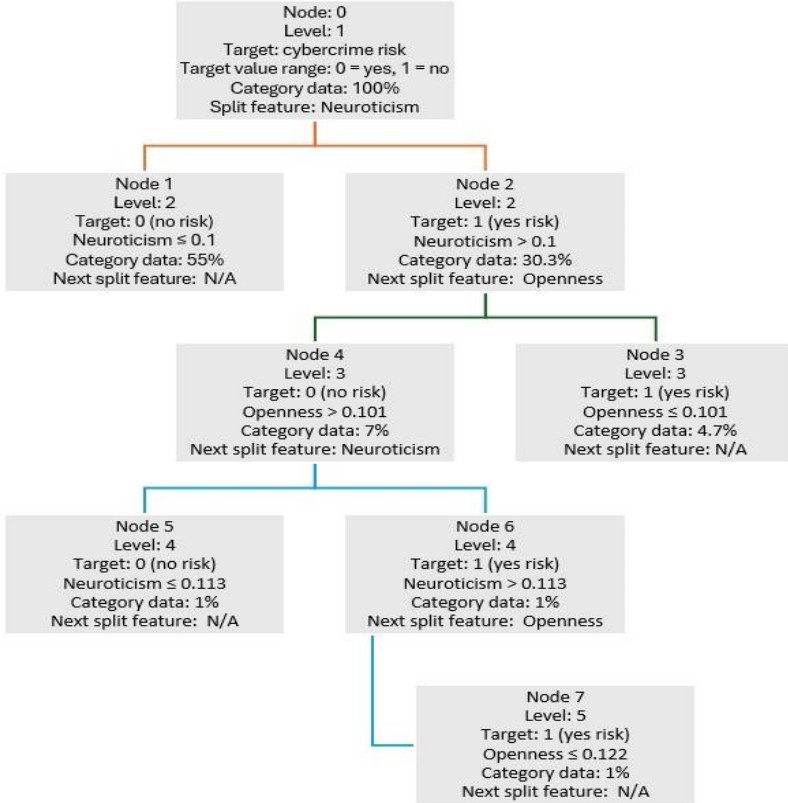

**Figure 4.** Learning tree diagram of cybercrime risk from employee behavior big data.

In the tree diagram (Figure 4), neuroticism was selected by the ML CHAID tree technique to make the first breakdown of the big data stream instead of the other 4 FFPT features—openness, conscientiousness, agreeableness, or extroversion. The root node 0 at level 1 on the top indicates neuroticism was the feature for the first splitting. The next level 2 represents the first split, to a leaf on the left and to another branch on the right. The first leaf at the left on level 2 in Figure 3 was based on low neuroticism (coefficients $\leq 0.1$), which was not associated with cyber risk, which captured 55% of the employee behavior big data. Adjacent to this at level 2 was the node 2 branch on the right, based on a high neuroticism coefficient > 0.1, which captured 30.3% of the big data.

At level 3, node 3, a lower level of openness $\leq 0.101$ was classified as a cyber risk, which represented 4.7% of the big data on employee behavior. On the same level 3, there was node 4, where higher levels of openness > 0.101 were associated with no cybercrime risk, which captured 7% of the employee behavior big data. Node 5 on the next level 4 was split on neuroticism $\leq 0.113$, indicating no cybercrime risk, which represented 1% of the data. Adjacent to this on level 4, node 6 split on neuroticism > 0.113, which was considered cybercrime risk and captured a further 1% of the employee behavior big data. Node 7 captured the remaining 1% of the employee behavior big data, with openness $\leq 0.122$ being classified as a cybercrime risk.

In summary, the neuroticism personality feature was the most important for most of the learning tree splits in Figure 4, with openness assisting with lower-level splits. of the remaining employee behavior big data. To interpret this learning tree model, we must keep in mind that cybercrime risk was determined by the keywords found in the big data stream, namely the use of URLs deemed risky based on the literature review. In other words, to be classified as a cybercrime risk, the employees may have embedded risky URLs while writing phrases typical of the FFPT neuroticism or openness personality factors.

The first theoretical interpretation of the learning tree analysis was that higher levels of neuroticism were usually associated with probable cybercrime risk and vice versa. That finding tends to support the extant literature, as well as the arguments by Lickiewicz [8],

that higher neuroticism would be associated with more cybercriminal-type behavior. Paradoxically, the most interesting interpretation was that once neuroticism was considered, then a lower openness coefficient became associated with cybercrime risk, which was contrary to the personality theory body of knowledge in the cybersecurity domain and contrary to what Lickiewicz [8] proposed. Overall, we could interpret the tree diagram in Figure 4 as suggesting that higher levels of FFPT neuroticism but lower levels of openness to new experiences tend to result in higher cybercrime risk employee behavior in big data.

### 3.4. Cybercrime Risk Feature Scatter Plot Analysis

Figure 5 is a four-dimensional scatter plot created to further explore how neuroticism and openness features impacted cybercrime risk. The learning tree CHAID analysis and diagram indicated how the big data of employee behavior could be grouped with the chosen features of neuroticism and openness, while this scatter plot will show how the data values contrast with one another when considering cybercrime risk. Due to the data's size, a condensed distribution was created so summary estimates could be plotted to fit into the display space. As explained in the methods, the summarized coefficient values for neuroticism were scaled on the *x*-axis, and openness on the *y*-axis.

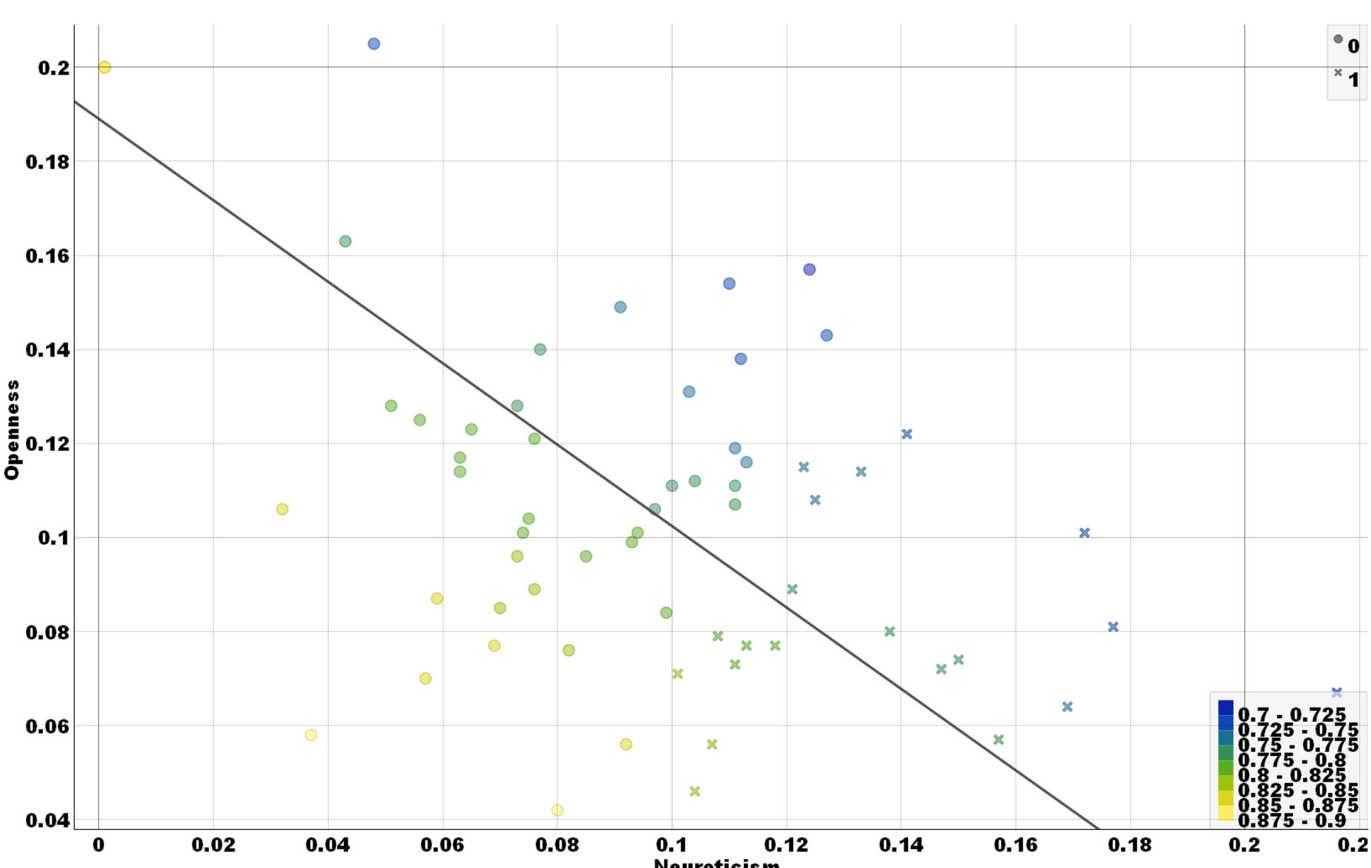

**Figure 5.** Scatter plot of FFPT neuroticism and openness by coefficient score and cybercrime risk.

As explained in the methods, three additional control attributes were added to the scatter plot: color shades, different symbols, and a slope line. The color was scaled to match the coefficient score of the cybercrime risk association (lighter yellow shades represented low-risk coefficients, and darker shades of blue represented higher risk coefficients). The symbol was arranged to differentiate data values between no cyber risk—shown as a '0'—and a potential cyber risk—shown as an 'x'. A regression slope line was superimposed in Figure 5.

It is clear from looking at the Figure 5 scatter plot that the cybercrime risk behaviors were associated with high neuroticism and low openness coefficient scores, towards the lower right. The regression line slope corroborates this interpretation that employee behavior in the big data stream can be categorized as potentially having a higher cybercrime risk if the neuroticism score is higher and the openness coefficient is lower. The regression slope line also suggests a theoretical split in the big data. One group is below the slope line, shown with yellow circles, representing high openness but no cybercrime risk, gradually evolving into a higher cyber risk (left to right). Similarly, the second group on top of the slope line contains higher openness and higher neuroticism coefficient scores.

*3.5. Cybercrime Risk Feature Heat Map Analysis*

The CHAID learning tree coefficient scores can be analyzed in a different manner than the scatter plot by using a heatmap to group together the more important FFPT features using cluster analysis. Figure 6 is a heatmap showing the top three FFPT features. Figure 6 includes neuroticism and openness, as earlier discussed in the tree diagram, but also agreeableness, since it was detected using other ML techniques as potentially having a negligent impact on cybercrime risk. No other features were strong enough to be detected with any ML technique.

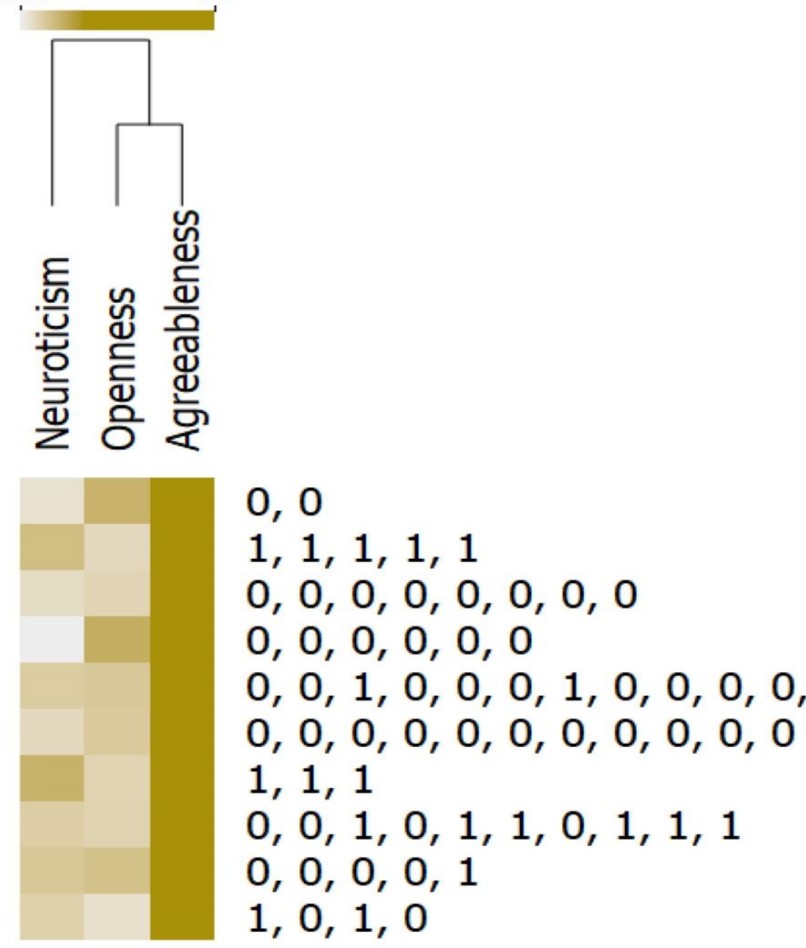

**Figure 6.** Heatmap of cybercrime risk (0 = no, 1 = yes) clustered by FFPT personality feature.

In Figure 6 heatmap, we can see the darker shades highlight strong or higher coefficients cross-referenced to the FFPT feature. The darkness scale ranges from 0 to 1, as shown in the legend at the top of Figure 6. The scale for the legend was derived from the FFPT feature coefficient scores. In this scale, a low value close to 0.001 indicates a small

alignment in the big data employee behavior to the FFPT feature, while a value close to 0.9 suggests a strong association of employee behavior to the feature. The top of Figure 6 includes a dendrogram visually illustrating the feature clustering relationship.

The values on the right side of Figure 6 contain the no cybercrime risk (0) versus cybercrime risk (1) findings. For example, we can see in Figure 5, on the right, that there are two row clusters with many 1 scores ('1,1,1' and '1,1,1,1,1'), referring to cybercrime risk. These two clusters with many 1's estimate on the right are colored in a darker shade on the left under the neuroticism column, with light colors under the openness feature. The repetitive '1' sections of the heatmap in Figure 6 contrast with the repeated '0' regions representing no cybercrime risk behavior detected, where the within-cluster counts ranged from six values of '111111' up to '0' features in one cluster.

A different FFPT factor, agreeableness, was colored dark in much the same shade for all the cybercrime risk clusters. This can be interpreted as most employee behavior aligning with an elevated level of the agreeable attribute in personality theory, but it was not associated with cybercrime risk likelihood. This finding also indicates that these cybercrime risk features were not significantly different across the big data. Therefore, agreeableness is not a significant FFPT factor in the current study results. We can also observe from the dendrogram at the top of Figure 6 that openness and agreeableness are closely related, while neuroticism is distinct in the big data.

## 4. Conclusions

The current study was successful—the RQ was answered. Additionally, a new cognitive computing paradigm was introduced. The FFPT construct from the psychology discipline was integrated into ML using the CHAID learning tree analysis technique to identify cybercrime risk from retrospective employee behavior big data. The results showed that two personality theory factors were associated with cybercrime risk, namely, neuroticism and openness. Specifically, higher employee neuroticism coefficients and lower openness scores were associated with greater organizational cybercrime risk. None of the other FFPT personality factors (agreeableness, conscientiousness, and extroversion) had any informative association with cybercrime risk.

A novel interdisciplinary paradigm for big data cognitive computing practice was demonstrated by integrating a proven theory from the psychology literature and leveraging ML best practices from other studies to analyze human behavior using a retrospective big data collection approach that was efficient, reliable, and valid. Over 8 GB of data in a data lake containing big data on employee behavior were analyzed with ML in a supercomputing Python environment. This was much faster, more valid, and more reliable as compared to traditional data collection methods such as surveys or interviews.

The results were valid, reliable, and credible, according to ML practice benchmarks. The cybercrime risk learning tree training model classification accuracy was 0.879, the estimated MAPE (error rate) was 12.1%, the recall-sensitivity indicator was 88%, the score was 0.879, the specificity metric was 83.4%, and the F1 overall quality score was 88%. Although the area under the receiver operating characteristic (AROC) coefficient was 0.879, the scatter plot curve of the training model versus a null model, incorporating the tradeoff between recall-sensitivity versus the 1-specificity value, was 67%, which was slightly lower than the goal of 70%, but it was acceptable in the context. The above coefficients, when evaluated together, illustrated that the study results were valid.

Reliability was established in two ways. First, the size of the big data provided an excellent context for rigorous training and testing of the ML CHAID learning tree model. The data was stable because it was retrospective, and it encompassed actual employee behavior for approximately one year. Second, the method for testing the training model was considered exceptional by ML practice standards. The n-fold approach was used, not with the strong 10-fold setting, but with a 20-fold resampling without replacement design. The large, big data size allowed for rigorous resampling.

Credibility was developed through the sampling technique and by articulating the research design. The intended population was identified as Fintech companies with a cybercrime risk (with a single case study selected as the sample). This sample was representative of the intended population because it was in a well-known Fintech district of New York State (USA), it was a multinational corporation with offices in several countries around the world, and at least one competitor had recently experienced a ransomware cyberattack. Therefore, the population sampling frame approach could be considered ideal for the current study. This allows the results to be generalized as having implications for other Fintech industry companies worldwide, at least in western-culture nations (because the case study organization was a multinational with offices in North America, Europe, and Asia). The author also argues that these results would generalize to other industries outside Fintech because the nature of employee behavior on corporate intranets should be homogeneous in business contexts representative of western-type cultures (e.g., in manufacturing, services, etc.).

Moreso, credibility was achieved by describing the research design and customized ML techniques in detail. The detail would be sufficient to allow another ML-experienced researcher to replicate and extend the current study. A conceptual research design workflow diagram was added to appease visual learning styles. Additionally, a sample dataset was provided (and authorized by the case study company) to provide insight and a reference template to assist future researchers.

### 4.1. Literature Contrasts, Detailed Implications, and Generalizations

An interesting outcome of the current study was to demonstrate how previous state-of-the-art work has been leveraged and extended. The empirical work of Hiremath et al. [6] was leveraged to identify high-risk URLs in a company website and, in turn, identify cybercrime risk behavior in the big data.

The empirical work of Dalal et al. [9] provided examples and inspiration for implementing the CHAID learning tree analysis technique in ML. Their analysis was much more detailed from a computer science perspective because they generated their own simulated data (over 123,000 records) to simulate cyberattacks. They used their simulated data to train and test two types of models based on CHAID and SVM. This could be considered the ensemble method of ML training model validation, although other researchers [19,20] recommended using as many techniques as possible, such as linear regression and random forest, to achieve methical triangulation. We could argue that additional ML techniques would not provide any new or better insights from the big data of employee behavior because CHAID learning tree analytics are the best ML technique when a single dichotomous target variable is available with mixed data type features (either semi-supervised or supervised labels). Nevertheless, other researchers are encouraged to try other ML techniques in this research design.

The theoretical literature meta-analysis by Martineau et al. [5] provided a relevant starting point for designing the criminal behavior analysis construct adapted for the current study. The theoretical work of Lickiewicz [8] was leveraged to extend those concepts developed by Martineau et al. [5] to identify the most relevant *a priori* personality theory for discovering cybercrime risk, the five-factor model from the psychology discipline. Their papers proposed models using literature reviews, but they did not test their constructs, so the current paper has extended their work and tested their ideas on big employee behavior. The implication is that the current design can be replicated using *a priori* theories from any discipline integrated into ML to analyze big data—that is a novel research design paradigm. Additionally, there is an implication of the research design concept for other *a priori* facets from other disciplines beyond human behavior, in particular educational psychology and criminology. The current study now opens the door to integrating any *priori* theory into ML and how to adapt or customize the algorithms to analyze big data. Consider this as a new paradigm inspiring abductive, deductive, and inductive ML big data research.

Another valuable outcome from the current study was the use of ML to collect actual employee behavior big data from a company intranet. This data collection technique is argued to be superior to asking participants to self-report their behavioral tendencies using surveys or interviews. It is logical to assert that the more intelligent an alleged cybercriminal is, then the more likely he/she will be able to circumvent even the most cleverly designed survey, which may employ social desirability and honesty-check items—even when reverse-coded into the instrument. Using ML to access employee behavior big data might be comparable to observation as a data collection method. While both the observation and the ML big data access techniques stand an excellent chance of collecting authentic data, a researcher can appreciate that observation is a more time consuming and intrusive method to achieve the same end. Observation will only be practical to collect a small sample size. By comparison, ML can access trillions, e.g., petabytes, of constantly changing big data. Thus, we could generalize that the current study has provided valuable information for the scholar community of practice, namely cognitive computing ML applied to big data, and the specific cybercrime risk findings constitute a preliminary starting point for stakeholders.

We can argue that ML algorithms have advantages over traditional multivariate parametric or nonparametric statistical techniques because any ML program has no underlying distribution assumptions and big data can be analyzed. You cannot analyze big data with modern statistical applications, nor can you implement true conditional logic. ML can also process dynamic big data with appropriate checks for reliability, validity, and credibility. No statistical program in today's world can achieve such a task.

The implications of the current study should be uniquely interesting to the business and managerial schools of thought. We could argue that organizational technology decision-makers or managers of any sector could use ML models to identify employees with cybercrime risk behaviors. Once those employees are identified, we could assist them by providing them with cybersecurity policy refresher training, mentoring, and coaching. The research design can also be generalized very widely.

These results should provide relevant implications of interest to all stakeholders, including researchers, practitioners, and executive decision-makers. Some concepts from this study ought to be of interest to everyone around the world. For example, it would be valuable to have a cybercriminal risk model in any context. Our current society is very dependent on technology, and this is what cybercriminals attack. Smartphones are used everywhere. What if a programmer created a cyber-ransomware virus for a popular smartphone application that was installed by trillions of people worldwide? Is such a phenomenon possible? How would that risk propagate when one considers that company executives and employees use their smartphones at the workplace as well as for off-site work? The implications of the current study could be transformed with imagination to generalize as a new global threat: a cyber-pandemic.

*4.2. Limitations and Caveats*

There were several important limitations to share in the current study. First, the big data source was collected from one large company as a single case study, based in the USA with several overseas offices. While we can attest to the reliability, validity, and credibility of the current study (as discussed above), a case study is still a sample, and a sample is not the actual population. More data would have to be collected and the model replicated to confirm its reliability, validity, and credibility through replication.

Second, the big data population was from a western-oriented culture, the USA. Also, most of the global psychology discipline literature, including the FFPT integrated into the ML for the current study, was developed in the USA. This puts a western culture slant on the population and the *a priori* constructs. Therefore, more studies need to be undertaken to replicate these concepts across different industries, different geographical cultures, and even using other theories beyond the psychology discipline. Additionally, the entire body of knowledge and data were in English; it would be interesting to replicate the current study in another popular language, such as Spanish.

Another limitation was that the big data was collected from a company intranet, which was entirely governed by Microsoft applications. Microsoft SharePoint was the intranet, mostly Microsoft applications were installed, and the Microsoft Python supercomputing environment was used for the ML environment. This presents a methodological bias. This bias could be examined and neutralized by replicating the current study's research design using another platform with a different case study. As hinted in the methods section, other vendors, including Google, provide comparable ML platform services, and once the current study is disseminated, it may help other researchers convince more companies to collaborate in this type of scholar research to advance the state-of-the-art and provide benefits to the business community.

Additionally, we do not know if AI within the Microsoft products may have confounded these results in some unknown manner. Microsoft has several AI initiatives, such as their large language model for SharePoint and their AI cognitive services utilized for the current study. We do not know how these products interacted with the big data, either before or during the study, without detection. Microsoft has begun to embed AI into many of its products [15], which could have altered employee behavior and outputs. A simplistic example of their AI interference could be to make an unintended keyword substitution through automatic spell-check functions on SharePoint as well as on smartphones. Additional examples of AI impact could be duplication of keywords when an employee's outputs are automatically distributed through rules or replicated by other staff. The impact of AI on company SharePoint big data sources should be investigated in future studies.

The validity and reliability of the data may have had an impact on the results. We do not know the risk of data mortality, as some employees could have deleted their posts before the big data collection, some employees could have been terminated and had their data removed, and some data may have been unreadable due to encryption. This raises another limitation in that some data could look like an employee created it, but that may not have been the case (e.g., paraphrasing or copy-pasting), so the big data could have inadvertently caused false positives or false negatives. Furthermore, it was not clear to what extent the employees used their smartphones or other equipment, in the workplace or off-site, while accessing the company intranet. Future studies could investigate the use of big data to control the types of employee technology and access points.

### *4.3. Future Study Recommendations*

In closing, it is important to note that these results are exploratory; the goal was to answer the RQ and introduce a novel cognitive computing paradigm. In the future, the ML ensemble approach could be used to achieve methodological triangulation, and more types of big data could be accessed, such as audio recordings, to achieve data triangulation. The author argues that the big data cognitive computing field is in its infancy, so much more research will be needed to fully develop this new paradigm.

**Funding:** No specific funding was received for this study.

**Institutional Review Board Statement:** The big data was provided after the case study company removed confidential and identifying information. The author signed a nondisclosure and confidentiality agreement with the case study company; a redacted copy of the authorization was provided to Assistant Editor on 8 January 2024.

**Informed Consent Statement:** All big data were retrospective—there were no participants, thus, informed consent was not applicable.

**Data Availability Statement:** Strang, K. D. 2023. Cybercrime Risk Keyword Sample Authorized by Case Study Company, Harvard Dataverse: USA: https://doi.org/10.7910/DVN/FPDLOR (accessed on 1 March 2024).

**Acknowledgments:** The author thanks the four double-blind peer reviewers for taking significant effort in providing very useful constructive feedback. The author thanks Bulcsu Szekely, LUT

University, School of Engineering Science, for his informal peer-reviews prior to submission. The author thanks Assistant Editor for moderating four labor-intensive revision cycles and proofing.

**Conflicts of Interest:** The author declares no conflicts of interest.

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
