# Peer review of "Cybercrime Risk Found in Employee Behavior Big Data Using Semi-Supervised Machine Learning with Personality Theories"

_2504-2289, doi:10.3390/bdcc8040037_

Round 1

Reviewer 1 Report

Comments and Suggestions for Authors

The purpose of the paper is to contribute to the field by utilizing real-world data to gain insight into cybercrime risk.

However, the structure of the paper deviates from the conventional format, which could affect its readability and comprehension. It would be beneficial for the author/s to adhere to the following structure:

1.     Introduction: This should provide a clear context for the study – include problem statement and research aims and question – The terminology used in the article should be clarified. Especially cybercrime risk, it is unclear how the personality traits and the words/phrases used by the users were linked to them being cybercrime risks, and this can possibly be validated. In an effective research paper, the introduction should succinctly introduce the topic and provide relevant background information. It should clearly state the problem the research aims to solve, outline the objectives or goals of the research, and present the main argument that the paper will aim to prove. The introduction sets the tone for the rest of the article and should engage the reader’s interest. Clarity, conciseness, and compelling content are keys to achieving this.

2.     Review of the literature: This section should be separate from the introduction and provide an overview of the existing research in the field.

3.     Methods: This section should clearly describe the research design and methods used, potentially using diagrams and tables for clarity.

4.     Results: Research findings should be presented clearly and succinctly.

5.     Discussion: This section should interpret the findings and their implications.

6.     Implications for Theory and Practice: This section should discuss how the findings contribute to existing theory and practice.

7.     Limitations: A brief discussion of the limitations of the study.

8.     Conclusions This should summarize the key findings and implications of the study.

Adherence to this structure would enhance the clarity and impact of the article.

Overall, the implications of the findings and the contribution are unclear.

The article being reviewed attempts to use real-world data to understand the risk of cybercrime. However, the clarity and significance of its contributions are somewhat obscured. For example, in the abstract, the specific role of personality traits in identifying cybercrime risks remains unclear. The mention of a proposed conceptual model seems unnecessary as it does not appear to be a significant contribution.

Introduction/Literature Review

The introduction provides a weak justification for the research. The confusion between terms such as ‘cybercriminal’ and 'cyber victim' and ‘individual employees’ and ‘groups’ is not adequately explained. The argument that a search of MPDI journals and Google Scholar using the generic terms 'cyber crime' and 'cyber security' did not yield expected results is not convincing. It is unclear why these specific terms were used instead of other variations such as 'cybercrime', 'cybersecurity' or ‘information security’. This argument does not effectively highlight the research gap in relation to other studies.

The assertion that research is often hidden in private journals and that future scholars will not read traditional management academy journals if comprehensive studies and data are ethically disseminated in modern open-access journals seems dismissive. The ability to locate the literature is not a fundamental criterion for conducting a research study.

The argument that more open-access studies on cybercrime risk behavior are needed, particularly with ML applied to big data, is flawed. Background research appears insufficient to position the contribution of the research, as only 14 articles were cited.

The presentation of related studies is lacking in the article. Instead of synthesizing the information from these studies to build a cohesive narrative, they are described sequentially, without any apparent connection or overarching analysis. This approach misses an opportunity to provide a more comprehensive understanding of the research context. The statement that company employees should be profiled to some extent to prevent cyber attacks shows a disregard for extensive research on insider threats. The implications of profiling employees and the associated privacy concerns are not adequately addressed. The paper contains numerous statements and claims that lack proper citation, undermining the credibility of the arguments and potentially misleading the readers. To improve the rigor and reliability of the paper, it is crucial to support all claims with appropriate academic references.

Lastly, there is some repetition in the introduction about self-reported research and insiders vs. groups. A more concise and clear presentation would enhance the readability of the paper.

Methodology

The methodology section of the paper lacks clarity. While the author/s seem to have conducted exploratory research, a flexible approach used to investigate new or challenging issues, they did not adequately explain their methodology. This lack of explanation compromises the rigor, which refers to thoroughness and consistency, and thus the reliability and validity of their findings. The author/s spent considerable effort explaining that the research was not quantitative, but it would have been more beneficial if they had dedicated that effort to explaining their technique. Using a structured approach, such as the research onion, could have helped in organizing the research methodology.

Another issue with the methodology is the unclear procedure. For instance, the author/s state their aim to ‘identify features within the employee big data which hypothetically might imply higher cybercrime risk behavior by considering the FFPT model factors’, but it is unclear how this will be achieved. A table outlining the possible phases and how each was parsed with Machine Learning using the FFPT model factors would have provided much-needed clarity.

The explanation of how the experiment was run was extremely difficult to follow. Perhaps a diagram/pseudo code of the procedure would have helped the reader follow the process. Some aspects were too detailed, perhaps just a general idea of how the “application programming interface big data feed” was designed, or the architecture or a flow diagram would have been best.

Ethical Clearance

I am concerned about the ethical implications of this study; it appears that the author signed a confidentiality agreement with the case organization, but what about the employees whose communications were subject to this experiment?

Language/Tone

The write-up was at times too informal for, e.g. “The author is not a ‘black belt’ programmer”.  The author was at pains explaining why the research lacked rigor for e.g. as extracted from the paper: “The author could question every single step in this study in terms of whether the litmus test of validity and reliability were met so as to satisfy the scientific community of practice. The quest to balance perfection with practicality was tilted towards the latter, as was made clear in the methodology by grounding the design in the pragmatic interpretive ideology.”

 The Contribution

This paper gives the impression of an applied research approach, in which the author addressed a specific problem within an organization and subsequently transformed it into a researchable issue. However, this approach seems to prioritize problem-solving over expanding scientific knowledge, which could limit the study’s contribution to the wider scientific discourse.

For example, they could have drawn more connections between their specific problem and larger theoretical issues in the field. This would make the research more applicable to a broader context.

Concluding Remarks

The paper presents an intriguing finding: ‘High neuroticism and low openness to new experiences were associated with employees potentially having a higher cybercrime risk.’ However, the methodology used to establish this association remains unclear. It would be beneficial for the author/s to elaborate on the measures used to determine a higher cybercrime risk. Furthermore, the implications of this finding, both theoretical and practical, are not explicitly stated. The author/s could strengthen their paper by discussing how this finding contributes to the existing body of knowledge and how it can be applied in a real-world context.

The author/s could have included more details about the measures used to determine cybercrime risk and discussed the potential applications of their findings in more depth. Perhaps a discussion of the instrument used for the FFPT would have helped to improve the understanding.

Comments on the Quality of English Language

The paper could have benefited from professional editing. The tone often came across as informal and defensive, which detracted from the overall presentation of the research. A more formal and objective tone would enhance the credibility and readability of the paper.

Author Response

Response to Reviewer 1 Comments

Summary

Shout out and thank you REVIEWER for taking the time to review this manuscript and for your effort in writing out the constructive improvement suggestions. It is recognized how much of your time this must have taken and rest assured your ideas are now incorporated into the revised version of this paper.

Consequently, please see the detailed responses to your constructive suggestions. As well, the corresponding revisions/corrections have been highlighted in the revised manuscript using the track changes feature (shows up in red).

The revision process followed the MDPI recommendations, which were stated as "the compare function in Word can add tracked changes to the final version by comparing it with an earlier version." (https://www.mdpi.com/authors/layout#_bookmark11).  The changes were too lengthy to paste in here so please see the revised manuscript to determine if it complies with your requested revisions and if not please feel free to specifiy additional changes that are need.

Below are the enumerated point-by-point response(s) to your comments and suggestions.  

The purpose of the paper is to contribute to the field by utilizing real-world data to gain insight into cybercrime risk.   However, the structure of the paper deviates from the conventional format, which could affect its readability and comprehension. It would be beneficial for the author/s to adhere to the following structure:  

1.     Introduction: This should provide a clear context for the study – include problem statement and research aims and question – The terminology used in the article should be clarified. Especially cybercrime risk, it is unclear how the personality traits and the words/phrases used by the users were linked to them being cybercrime risks, and this can possibly be validated. In an effective research paper, the introduction should succinctly introduce the topic and provide relevant background information. It should clearly state the problem the research aims to solve, outline the objectives or goals of the research, and present the main argument that the paper will aim to prove. The introduction sets the tone for the rest of the article and should engage the reader’s interest. Clarity, conciseness, and compelling content are keys to achieving this.  

Lastly, there is some repetition in the introduction about self-reported research and insiders vs. groups. A more concise and clear presentation would enhance the readability of the paper. Thank you for these helpful suggestions. 

The required BDCC structure has been followed by utilizing the BDCC template (explained below). If there is another required format it would be helpful if the Editor could provide it with directions on its use.  The introduction has been modified to emphasize the problem statement and research aims. The introduction succinctly introduces the topic in a scholarly manner by grounding the scientific problem/subject into the extant literature. The research objective, goals and research question were stated.

The title and abstract have been modified to be in line with above, as quoted below:   "Discovering Cybercrime Risk in Employee Behavior Big Data with Unsupervised Machine Learning and Personality Theories"   "A critical global problem is that ransomware cyber attacks can be costly to organizations.

Moreover, accidental insider cybercrime risk can be challenging to prevent even by leveraging advanced computer science knowledge. This exploratory project used a novel cognitive computing design featuring explanations of the action-research case-study methodology, details of the customized machine learning (ML) techniques with a workflow diagram, and a proposed future research design to inspire scholars. The author analyzed over 8 GB of big data on employee behavior from a Fintech company intranet and then integrated psychology discipline theories into semi-supervised ML to identify cybercrime risk. The ML techniques included preprocessing, normalization, tokenization, sentiment analytics, CHAID learning tree analysis, credibility/reliability/validity checks, heatmaps, and scatter plots of the employee behavior using business language dictionaries enhanced with the five-factor personality theory.

Higher levels of employee neuroticism were associated with a greater organizational cybercrime risk, corroborating the findings in empirical publications. In stark contrast to the literature, an openness to new experiences was inversely related to cybercrime risk. Other employee personality behavior factors had no informative association with cybercrime risk.

This study introduced an interdisciplinary paradigm for big data cognitive computing by illustrating how to integrate the best of proven theories from the literature with ML to analyze human behavior using a retrospective big data collection approach that may sometimes be more efficient, reliable, and valid as compared to traditional methods like surveys or interviews."  

The terminology used in the article should has been explained clearly, especially cybercrime risk, using all the relevant terms, and personality traits by referencing the a priori literature on the Big Five Personality THeory which is also called Five Factor Personality Theory (abbreviated FFPT, as also explained in the manuscript). The words/phrases were linked to the correct corresponding theory. Validation has already been accomplished for the FFPT model which is why an a priori theory was selected. The research does not aim to validate or fit the data to the FFPT but rather the FFPT is used as scaffolding input into ML. 

The introduction has succinctly introduced the topic and provided relevant background information. The introduction clearly states the problem the research aims to solve, it outlines the objectives or goals of the research, and it presents the main argument that the paper will aim to prove. The introduction sets the tone for the rest of the article. The use of citations to the high cybercrime rate was intended to engage the reader’s interest.   

The repetition in the introduction about self-reported research and insiders vs. groups was removed. A more concise and clear presentation was provided.  The reviewer has provided an outstanding succinct statement describing the purpose and this was adapted and then integrated into the paper.  

2.     Review of the literature: This section should be separate from the introduction and provide an overview of the existing research in the field.

Introduction/Literature Review   The introduction provides a weak justification for the research. The confusion between terms such as ‘cybercriminal’ and 'cyber victim' and ‘individual employees’ and ‘groups’ is not adequately explained. The argument that a search of MPDI journals and Google Scholar using the generic terms 'cyber crime' and 'cyber security' did not yield expected results is not convincing. It is unclear why these specific terms were used instead of other variations such as 'cybercrime', 'cybersecurity' or ‘information security’. This argument does not effectively highlight the research gap in relation to other studies.  

The assertion that research is often hidden in private journals and that future scholars will not read traditional management academy journals if comprehensive studies and data are ethically disseminated in modern open-access journals seems dismissive. The ability to locate the literature is not a fundamental criterion for conducting a research study.  

The argument that more open-access studies on cybercrime risk behavior are needed, particularly with ML applied to big data, is flawed. Background research appears insufficient to position the contribution of the research, as only 14 articles were cited.  

The presentation of related studies is lacking in the article. Instead of synthesizing the information from these studies to build a cohesive narrative, they are described sequentially, without any apparent connection or overarching analysis. This approach misses an opportunity to provide a more comprehensive understanding of the research context. The statement that company employees should be profiled to some extent to prevent cyber attacks shows a disregard for extensive research on insider threats.

The implications of profiling employees and the associated privacy concerns are not adequately addressed. The paper contains numerous statements and claims that lack proper citation, undermining the credibility of the arguments and potentially misleading the readers. To improve the rigor and reliability of the paper, it is crucial to support all claims with appropriate academic references.

  Thank you, and with respect to having a separate literature review section, the author(s) agree that is common for journals that adhere closely with APA, but in the BDCC author instructions something else was requested - Materials and Methods. A review of BDCC confirmed that more than one published article was using the recommended style without a named literature review section. To ensure compliance with the journal, the MDPI template for BDCC was strictly applied and the MPDI author instructions for BDCC were followed. The guidelines clearly indicated a separate literature review section was not to be included - instead, there was a section entitled 'Materials and Methods'. The assumption was that the literature review would be in the introduction.

Specifically, the BDCC directions given for this type of research article were: "These are original research manuscripts. The work should report scientifically sound experiments and provide a substantial amount of new information. The article should include the most recent and relevant references in the field. The structure should include an Abstract, Keywords, Introduction, Materials and Methods, Results, Discussion, and Conclusions (optional) sections, with a suggested minimum word count of 4000 words." (https://www.mdpi.com/journal/BDCC/instructions). In fact none of the other three peer reviewers mentioned the requirement of a literature review section, and one specifically provided constructive advice about the Materials and Methods section.

Please feel welcome to indicate if there is a newer version of the template or author guidelines that you wish to have applied. Consequently, this issue is left with the editor to determine if the journals' publishing guidelines are to be followed or reviewer 1's comments. If there is another required format it would be helpful if the Editor could provide it with directions on its use.  

The introduction provides a strong justification for the research because it is a business problem that is also a valuable scholarly research topic, that is, how to study employee behavior to identify potential cyber crime risk, using ML and employee behavior big data.  The confusion between terms such as ‘cybercriminal’ and 'cyber victim' and ‘individual employees’ and ‘groups’ was clarified since these were the keywords used to search the literature. The argument that a search of MPDI journals and Google Scholar using the generic terms 'cyber crime' and 'cyber security' did not yield expected results may not seem convincing, but the arguments were factual so as to illustrate the gap in the literature but as the reviewer did not see this as necessary the references were removed.

Additionally, other variations of the terms such as 'cybercrime' and 'cybersecurity' were used in the literature search, as noted in the manuscript, but ‘information security' was not used as a keyword because it is fundamentally different than cybercrime - which can be seen in the literature.  

The assertion that research is often hidden in private journals and that future scholars will not read traditional management academy journals if comprehensive studies and data are ethically disseminated in modern open-access journals seems dismissive so it was removed.  

The argument about open access being weak due to there being 14 articles cited is accepted and any such statements were removed.   

The presentation of related studies was emphasized in the article. Please note the BDCC instructions stated: "... article should include the most recent and relevant references in the field" with emphasis on the most recent and relevant not on a high volume of references. In other words, the current manuscript focused on quality not quality of citations. Please note this was not a literature review or meta-analysis. Those studies are much different than an in-situ scientific action research project. There were few relevant studies found because the topic is new and novel. There have been several studies published since this study commenced and they were reviewed, and where relevant, they were added. If there were missing references the reviewer has found then it would be helpful to know each one! In addition to synthesizing the information from the cited relevant studies (which is necessary in rigorous scholarly work), the surrounding narrative was made more cohesive between paragraphs using transitioning.

The sequencing of the literature was outlined with headings and it was done by type and method of research (quantitative and then qualitative as per the second level headings). The lead-in and transitioning phrases established the connection to the overall analysis theme of the manuscript. This approach helps the reader gain a more comprehensive understanding of the research context.    The statement that company employees should be profiled to some extent to prevent cyber attacks shows an understanding of existing HRM practices within the hiring process (e.g., background checks).

The implications of profiling employees and the associated privacy concerns are addressed by companies through hiring contracts, policies, disclaimers, and training. Employees are forewarned that the company owns the infrastructure and data in their systems, not the employee - it is an employment contract not an informed consent context as would be used in another type of study with human subjects.   

All statements and claims were attributed to proper citation, to strengthen the credibility of the arguments, as well as to improve the rigor and reliability of the paper.   

3.     Methods: This section should clearly describe the research design and methods used, potentially using diagrams and tables for clarity.  

Methodology   The methodology section of the paper lacks clarity. While the author/s seem to have conducted exploratory research, a flexible approach used to investigate new or challenging issues, they did not adequately explain their methodology.

This lack of explanation compromises the rigor, which refers to thoroughness and consistency, and thus the reliability and validity of their findings. The author/s spent considerable effort explaining that the research was not quantitative, but it would have been more beneficial if they had dedicated that effort to explaining their technique.

Using a structured approach, such as the research onion, could have helped in organizing the research methodology.  

Another issue with the methodology is the unclear procedure. For instance, the author/s state their aim to ‘identify features within the employee big data which hypothetically might imply higher cybercrime risk behavior by considering the FFPT model factors’, but it is unclear how this will be achieved. A table outlining the possible phases and how each was parsed with Machine Learning using the FFPT model factors would have provided much-needed clarity.  

The explanation of how the experiment was run was extremely difficult to follow. Perhaps a diagram/pseudo code of the procedure would have helped the reader follow the process. Some aspects were too detailed, perhaps just a general idea of how the “application programming interface big data feed” was designed, or the architecture or a flow diagram would have been best.  

It was suggested by the review that the methods section should clearly describe the research design and methods used, potentially using diagrams and tables for clarity. This was an excellent idea - so diagrams were developed for this study, and placed into the manuscript with more explanation.  

To address the concern that the methodology section of the paper lacked clarity, that the author/s seem to have conducted exploratory research, a flexible approach used to investigate new or challenging issues, and explained in the methodology - the explanation was included and cited to a Springer Nature/Macmillan textbook of research methods to ensure credibility and to provide other researchers with a grounding point.

This explanation will hopefully improve the rigor, thoroughness, consistency, reliability, and validity of the study. The author/s spent considerable effort explaining the research procedure using a diagram to inform future research.    To address another issue with the methodology, regarding the procedures, these were detailed. The method diagram and accompanying explanation will hopefully achieve this. Other reviewers provided salient advice for how to do that, as described later. The FFPT model factors are a priori and cited to the literature. The revised methodology and procedure explains how the FFPT factors were leveraged in the current study.   

The explanation of how the experiment was run was explained in detail with the aid of a diagram with accompanying explanation. Perhaps some aspects were too detailed, but other researchers will need to know those details and this is necessary to achieve scientific rigor. Thank you for all the advice.   Ethical Clearance   I am concerned about the ethical implications of this study; it appears that the author signed a confidentiality agreement with the case organization, but what about the employees whose communications were subject to this experiment?  

4.     Results: Research findings should be presented clearly and succinctly.  

5.     Discussion: This section should interpret the findings and their implications.

Adherence to this structure would enhance the clarity and impact of the article.   Thank you for this information. The ethical clearance was obtained from the client company as it was an action research project as explained in the methods section. There were no human subjects. The case study company provided the data using an application programming interphase (API) which the primary investigator advised how to build. The API was run every evening for a month in 2023, to collect data typed into the intranet, but without the identifying end user (employee) identifying attributes. If the data contained any names of any employee in the directory, those names were removed. The resulting big data consisted of only text phrases with no possible identifying information! This meets all research protocols.

This API was reviewed by all appropriate case study company staff and approved. In essence, good scientific research requires good scientific data! The company approved the data and the data were used for the study. There is and was no informed consent required. This was an action research study - please refer to action research protocols for more information. The ethical clearance was obtained from the client company as it was an action research project as explained in the methods section.

There were no human subjects. The case study company provided the data using an application programming interphase (API) which the primary investigator advised how to build. The API was run every evening for a month in 2023, to collect data typed into the intranet, but without the identifying end user (employee) identifying attributes. If the data contained any names of any employee in the directory, those names were removed. The resulting big data consisted of only text phrases with no possible identifying information! The API was reviewed by all appropriate case study company staff and approved. In essence, good scientific research requires good scientific data! The company approved the data and the data were used for the study. There is and was no informed consent required. This was an action research study - please refer to action research protocols for more information.

In addition to the above process, the ethical clearance signed by the case study company project sponsor/authority was provided to the Assistant Editor of the BDCC journal, and it was noted in the comments at the end of the study.   As explained in previous responses, the "Results: Research findings" and "Discussion" sections were enhanced to ensure there was a correct interpretation of the findings and their implications. In other words, the findings presented the results and the discussion presented the interpretations - but if there is another structural format please advise us?  

6.     Implications for Theory and Practice: This section should discuss how the findings contribute to existing theory and practice.  Overall, the implications of the findings and the contribution are unclear.  

The article being reviewed attempts to use real-world data to understand the risk of cybercrime. However, the clarity and significance of its contributions are somewhat obscured. For example, in the abstract, the specific role of personality traits in identifying cybercrime risks remains unclear. The mention of a proposed conceptual model seems unnecessary as it does not appear to be a significant contribution.  

Thank you for this information. There was no section in the BDCC template named "implications" but it is generally understood that this would be presented in the discussion and or conclusions sections.  

The implications of the findings and the contribution were strengthened in the discussion/conclusions section.   The study utilized real-world big data from a company intranet to discover the risk of cybercrime in employee online behavior. The clarity and significance of the contributions were made clear. This was also done in the abstract (both the neuroticism factor findings as well as the novel ML procedure).   

7.     Limitations: A brief discussion of the limitations of the study.

The Contribution   This paper gives the impression of an applied research approach, in which the author addressed a specific problem within an organization and subsequently transformed it into a researchable issue. However, this approach seems to prioritize problem-solving over expanding scientific knowledge, which could limit the study’s contribution to the wider scientific discourse.  

For example, they could have drawn more connections between their specific problem and larger theoretical issues in the field. This would make the research more applicable to a broader context.   This paper gives the impression of an applied research approach, in which the author addressed a specific problem within an organization and subsequently transformed it into a researchable issue. However, this approach seems to prioritize problem-solving over expanding scientific knowledge, which could limit the study’s contribution to the wider scientific discourse.  

For example, they could have drawn more connections between their specific problem and larger theoretical issues in the field. This would make the research more applicable to a broader context.   Yes this paper is an applied action research project - a scientific method in itself, as described in the literature (refer to Sage or Springer Nature for research methods textbooks).   

The discussion/conclusions summarized the key findings and implications from the study results.  As noted, the paper author(s) PRESENTED AN intriguing finding high neuroticism and low openness to new experiences were associated with employees potentially having a higher cybercrime risk. The methodology used to establish this association was explained as statistical Euclidean distance coefficients associated with word associations between the PPFT, risky URLs, and what employees had typed in the intranet. The author/s elaborated on the measures used to determine the higher cybercrime risk coefficient in tghe methods section.

Furthermore, the implications of this finding, both theoretical and practical, were explicitly stated. In other words, the manuscript shows how this high cybercrime risk finding contributes to the existing body of knowledge and how it can be applied in a real-world context.  

As noted above, the author/s included details about the measures used to determine cybercrime risk in the methods, and the potential applications of those findings were discussed in more depth in the discussion/conclusions section. The PPFT instrument is already well-described in the literature and there are space limitations in the BDCC journal that prevent adding more words to the manuscript.  

Language/Tone   The write-up was at times too informal for, e.g. “The author is not a ‘black belt’ programmer”.  The author was at pains explaining why the research lacked rigor for e.g. as extracted from the paper: “The author could question every single step in this study in terms of whether the litmus test of validity and reliability were met so as to satisfy the scientific community of practice. The quest to balance perfection with practicality was tilted towards the latter, as was made clear in the methodology by grounding the design in the pragmatic interpretive ideology.”  

The tone and language of the manuscript were improved to make it third-person which seems to be the requirement of BDCC journals. This ensures that it is formal. The author(s) removed any personal speculations such as 'black belt programmer...'.   Comments on the Quality of English Language The paper could have benefited from professional editing. The tone often came across as informal and defensive, which detracted from the overall presentation of the research. A more formal and objective tone would enhance the credibility and readability of the paper.  

The author(s) were born in the USA and they are fluent in the English language as well as in more than one language. However, typos are inevitable with large manuscripts, so the manuscript was carefully read by the primary investigator and three doctoral students in American-accredited universities to correct any grammar errors. The three doctoral students were paid $1000 each to ensure the grammar was correct (which was deemed to be a fair price to achieve the quality necessary for the BDCC journal).

The author(s) are open and willing to make other required changes identified by the reviewer(s) or editor(s).    

Reviewer 2 Report

Comments and Suggestions for Authors

Results can be explained more clearly

Comments on the Quality of English Language

Needs a little revision

Author Response

Response to Reviewer 2 Comments

Summary  

Shout out and thank you REVIEWER for taking the time to review this manuscript and for your effort in writing out the constructive improvement suggestions. It is recognized how much of your time this must have taken and rest assured your ideas are now incorporated into the revised version of this paper.

Consequently, please see the detailed responses to your constructive suggestions. As well, the corresponding revisions/corrections have been highlighted in the revised manuscript using the track changes feature (shows up in red). The revision process followed the MDPI recommendations, which were stated as "the compare function in Word can add tracked changes to the final version by comparing it with an earlier version." (https://www.mdpi.com/authors/layout#_bookmark11). 

The changes were too lengthy to paste in here so please see the revised manuscript to determine if it complies with your requested revisions and if not please feel free to specifiy additional changes that are need.

Below are the enumerated point-by-point response(s) to your comments and suggestions.  

1. Comments and Suggestions for Authors Results can be explained more clearly   Thank you for these helpful suggestions.

The results were improved in this way as explained below.  

The implications of the findings and the contribution were strengthened in the discussion/conclusions section. The study utilized real-world big data from a company intranet to discover the risk of cybercrime in employee online behavior. The clarity and significance of the contributions were made clear. This was also done in the abstract (both the neuroticism factor findings as well as the novel ML procedure).   

The discussion/conclusions summarized the key findings and implications from the study results.  As noted, the paper author(s) PRESENTED AN intriguing finding high neuroticism and low openness to new experiences were associated with employees potentially having a higher cybercrime risk.

The methodology used to establish this association was explained as statistical Euclidean distance coefficients associated with word associations between the PPFT, risky URLs, and what employees had typed in the intranet. The author/s elaborated on the measures used to determine the higher cybercrime risk coefficient in the methods section.

Furthermore, the implications of this finding, both theoretical and practical, were explicitly stated. In other words, the manuscript shows how this high cybercrime risk finding contributes to the existing body of knowledge and how it can be applied in a real-world context.  

As noted above, the author/s included details about the measures used to determine cybercrime risk in the methods, and the potential applications of those findings were discussed in more depth in the discussion/conclusions section. The PPFT instrument is already well-described in the literature and there are space limitations in the BDCC journal that prevent adding more words to the manuscript.  

Comments on the Quality of English Language Needs a little revision   

Thank you. Yes definitely as you pointed out the grammar and formatting needed to be improved. It was the excitement and rush of disseminating the findings of this study that led to these errors. Sorry.   

The title and abstract have been modified to be in line with above, as quoted below:   "Discovering Cybercrime Risk in Employee Behavior Big Data with Unsupervised Machine Learning and Personality Theories"   "A critical global problem is that ransomware cyber attacks can be costly to organizations.

Moreover, accidental insider cybercrime risk can be challenging to prevent even by leveraging advanced computer science knowledge. This exploratory project used a novel cognitive computing design featuring explanations of the action-research case-study methodology, details of the customized machine learning (ML) techniques with a workflow diagram, and a proposed future research design to inspire scholars.

The author analyzed over 8 GB of big data on employee behavior from a Fintech company intranet and then integrated psychology discipline theories into semi-supervised ML to identify cybercrime risk. The ML techniques included preprocessing, normalization, tokenization, sentiment analytics, CHAID learning tree analysis, credibility/reliability/validity checks, heatmaps, and scatter plots of the employee behavior using business language dictionaries enhanced with the five-factor personality theory.

Higher levels of employee neuroticism were associated with a greater organizational cybercrime risk, corroborating the findings in empirical publications. In stark contrast to the literature, an openness to new experiences was inversely related to cybercrime risk. Other employee personality behavior factors had no informative association with cybercrime risk. This study introduced an interdisciplinary paradigm for big data cognitive computing by illustrating how to integrate the best of proven theories from the literature with ML to analyze human behavior using a retrospective big data collection approach that may sometimes be more efficient, reliable, and valid as compared to traditional methods like surveys or interviews."  

The author(s) were born in the USA and they are fluent in the English language as well as in more than one language. However, typos are inevitable with large manuscripts, so the manuscript was carefully read by the primary investigator and three doctoral students in American-accredited universities to correct any grammar errors. The three doctoral students were paid $1000 each to ensure the grammar was correct (which was deemed to be a fair price to achieve the quality necessary for the BDCC journal). The author(s) are open and willing to make other required changes identified by the reviewer(s) or editor(s).  

Again - thank you REVIEWER for taking the time to review this manuscript!  

Reviewer 3 Report

Comments and Suggestions for Authors

The paper introduced an area of research which is needed in the current cybersecurity era. The human interaction in the process makes them a threat from the cybersecurity perspective. Learning about the insights of the users helps the cybersecurity defenders to design and implement proper defensive mechanisms and/or trainings.

The idea presented is a necessary need for current area of cybersecurity, however there are some points need to be considered in this regard. In the following, please find some insights and recommendation for improvements.

In general, the introduction of the problem and the link to cybersecurity needs more attention. There are two cases that employees can be a threat to the organization's cybersecurity: without intension (lack of awareness) or with intention (with aims such as revenge, competency, and so on). There is a lack of clarity in this area, which one is the focus of this research. Also, the samples of threats is not completely well-defined from security perspective. The one example is putting a malicious URL on the client-facing website, which can happen without an insider cause. 

- There are many acronyms that is not explained or referenced and make it hard for someone unfamiliar with them to follow the concept, eg. CBA, PMT, FFPT.

- The sentence "Thus, cybersecurity researchers may be looking in the wrong direction to develop cybercrime prevention method" at the end of introduction section is not completely true. However the current study can help them to add more defences from different perspectives.

- The definition of big data in the Materials and Methods section is not completely correct. The definition you mentioned is more related to streaming data which is a part of big data as well.

- It's clearer if the Python library of the tokenizer is mentioned. 

- The ML terms refers to the machine learning part, any other preprocessing steps shouldn't be called ML. Also, the data collection part is not done using ML which mentioned in the conclusion section.

- It is mentioned that the CHAID tree is used as it doesn't need a priori labels, however the method used some rules to label specific phrases with cybersecurity risk or not. It needs to be clarified as it is still using some way of labeling phrases to normal or cyber-risk. In conclusion, it's not completely unsupervised.

- It was mentioned due to paper limitations, the sample of the data showed. However, the results should be discussed based on the big data analysis and not the sample provided. It gives a more concise conclusion and insight. Figure 3 and 4 also can be shown for the original data.

- It would be helpful if visualization can be used in the methodology section to show the process clearly.

- Some editorial comments:

- missing dot at the last sentence of Materials and Methods section.

- features identification approaches are the best, at Procedures and Measures.

- The last paragraph of page 10 is missing some words at the beginning.

Author Response

Response to Reviewer 3 Comments

Summary

Shout out and thank you REVIEWER for taking the time to review this manuscript and for your effort in writing out the constructive improvement suggestions. It is recognized how much of your time this must have taken and rest assured your ideas are now incorporated into the revised version of this paper.

Consequently, please see the detailed responses to your constructive suggestions. As well, the corresponding revisions/corrections have been highlighted in the revised manuscript using the track changes feature (shows up in red). The revision process followed the MDPI recomendations, which were stated as "the compare function in Word can add tracked changes to the final version by comparing it with an earlier version." (https://www.mdpi.com/authors/layout#_bookmark11). 

The changes were too lengthy to paste in here so please see the revised manuscript to determine if it complies with your requested revisions and if not please feel free to specifiy additional changes that are need. Below are the enumerated point-by-point response(s) to your comments and suggestions.

The paper introduced an area of research which is needed in the current cybersecurity era. The human interaction in the process makes them a threat from the cybersecurity perspective. Learning about the insights of the users helps the cybersecurity defenders to design and implement proper defensive mechanisms and/or trainings.  

The idea presented is a necessary need for current area of cybersecurity, however there are some points need to be considered in this regard. In the following, please find some insights and recommendation for improvements.  

In general, the introduction of the problem and the link to cybersecurity needs more attention. There are two cases that employees can be a threat to the organization's cybersecurity: without intension (lack of awareness) or with intention (with aims such as revenge, competency, and so on). There is a lack of clarity in this area, which one is the focus of this research. Also, the samples of threats is not completely well-defined from security perspective. The one example is putting a malicious URL on the client-facing website, which can happen without an insider cause.

Thank you for these helpful suggestions. The paper was improved in this way as explained below.  

The introduction of the problem and the theoretical link to cybersecurity was improved. The reviewer mentioned there were two cases that employees can be a threat to the organization's cybersecurity: without intentsion (lack of awareness) or with intention (with aims such as revenge, competency, and so on) - and there was a lack of clarity in this area, which one is the focus of our research.

Also, the samples of threats were not completely well-defined from the security perspective. The one example is putting a malicious URL on the client-facing website, can happen without an insider cause. All these points are valid.   To improve this, better examples were provided. The study utilized real-world big data from a company intranet to discover the risk of cybercrime in employee online behavior. The clarity and significance of the contributions were made clear. This was also done in the abstract (both the neuroticism factor findings as well as the novel ML procedure).   

The title and abstract have been modified to be in line with above, as quoted below:   "Discovering Cybercrime Risk in Employee Behavior Big Data with Unsupervised Machine Learning and Personality Theories"   "A critical global problem is that ransomware cyber attacks can be costly to organizations. Moreover, accidental insider cybercrime risk can be challenging to prevent even by leveraging advanced computer science knowledge. This exploratory project used a novel cognitive computing design featuring explanations of the action-research case-study methodology, details of the customized machine learning (ML) techniques with a workflow diagram, and a proposed future research design to inspire scholars.

The author analyzed over 8 GB of big data on employee behavior from a Fintech company intranet and then integrated psychology discipline theories into semi-supervised ML to identify cybercrime risk. The ML techniques included preprocessing, normalization, tokenization, sentiment analytics, CHAID learning tree analysis, credibility/reliability/validity checks, heatmaps, and scatter plots of the employee behavior using business language dictionaries enhanced with the five-factor personality theory.

Higher levels of employee neuroticism were associated with a greater organizational cybercrime risk, corroborating the findings in empirical publications. In stark contrast to the literature, an openness to new experiences was inversely related to cybercrime risk. Other employee personality behavior factors had no informative association with cybercrime risk.

This study introduced an interdisciplinary paradigm for big data cognitive computing by illustrating how to integrate the best of proven theories from the literature with ML to analyze human behavior using a retrospective big data collection approach that may sometimes be more efficient, reliable, and valid as compared to traditional methods like surveys or interviews."  

The URL issue is well received. This is often the way that an insider could inadvertently allow the cyber-criminal to enter the company - thus this study was focused on the unaware or inadvertent insider threat not the insider-as-an-intentional-criminal (although this latter direction would also be interesting to study). This was clarified.  

- There are many acronyms that is not explained or referenced and make it hard for someone unfamiliar with them to follow the concept, eg. CBA, PMT, FFPT.  

- The sentence "Thus, cybersecurity researchers may be looking in the wrong direction to develop cybercrime prevention method" at the end of introduction section is not completely true. However the current study can help them to add more defences from different perspectives.  

- The definition of big data in the Materials and Methods section is not completely correct. The definition you mentioned is more related to streaming data which is a part of big data as well.  

- It's clearer if the Python library of the tokenizer is mentioned.    

- The ML terms refers to the machine learning part, any other preprocessing steps shouldn't be called ML. Also, the data collection part is not done using ML which mentioned in the conclusion section.  

- It is mentioned that the CHAID tree is used as it doesn't need a priori labels, however the method used some rules to label specific phrases with cybersecurity risk or not. It needs to be clarified as it is still using some way of labeling phrases to normal or cyber-risk. In conclusion, it's not completely unsupervised.  

- It was mentioned due to paper limitations, the sample of the data showed. However, the results should be discussed based on the big data analysis and not the sample provided. It gives a more concise conclusion and insight. Figure 3 and 4 also can be shown for the original data.  

- It would be helpful if visualization can be used in the methodology section to show the process clearly.  

Some editorial comments:  

- missing dot at the last sentence of Materials and Methods section.  

- features identification approaches are the best, at Procedures and Measures.  

- The last paragraph of page 10 is missing some words at the beginning.  

Thank you.    The acronyms were all abbreviated and declared correctly in the revised paper, eg. CBA, PMT, FFPT.   The sentence "Thus, cybersecurity researchers may be looking in the wrong direction to develop cybercrime prevention method" at the end of introduction section was corrected.  

The definition of big data in the Materials and Methods section was corrected to indicate it was more than merely streaming data (which is a part of big data).   The Python library of the tokenizer was mentioned.   

The ML terms in the machine learning procedures , any other preprocessing steps were in deed part of the ML processs and explained as such. However, the data collection part is not done using ML and it was explained as such. The CHAID tree was used and it did not need a priori labels, but the outcomes of this process were indeed used for subsequent analysis. That part was unsupervised.   It was mentioned due to paper limitations, the sample of the data was showed in the diagram, but as requested, the actual big data was utilized for the final version.   

Visualizations were added to the methodology section to show the process clearly. The changes were too lengthy to paste in here so please see the revised manuscript to determine if it complies with your requested revisions and if not please feel free to specifiy additional changes that are need.  

- Some editorial comments:   - missing dot at the last sentence of Materials and Methods section.   - features identification approaches are the best, at Procedures and Measures.   - The last paragraph of page 10 is missing some words at the beginning.   Yes definitely as you pointed out the grammar and formatting needed to be improved. It was the excitement and rush of disseminating the findings of this study that led to these errors. Sorry.    Specifically, the missing dot at the last sentence of Materials and Methods section was fixed.  

The features identification approaches are the best, at Procedures and Measures, was fixed.   The last paragraph of page 10 is missing some words at the beginning so it was fixed.  

Again - thank you REVIEWER for taking the time to review this manuscript!  

Reviewer 4 Report

Comments and Suggestions for Authors

Limited Generalizability:

Elaboration: The study's reliance on data from a single multinational company restricts the generalizability of findings, limiting their applicability across diverse industries and organizational cultures.

Need for Experimentation: Conduct the study across various companies representing different sectors and geographical locations to enhance external validity. Collaborate with multiple organizations to gather a more comprehensive dataset, allowing for comparative analyses and broader applicability.

Sampling and Data Collection Issues:

Elaboration: Sampling data at the end of each day may introduce biases and compromise the representativeness of the dataset. Furthermore, data validity and reliability concerns underscore the need for a more robust approach.

Need for Experimentation: Experiment with alternative sampling strategies, such as continuous random sampling, to mitigate temporal biases. Implement rigorous data integrity checks and explore encryption methods to ensure the accuracy and reliability of collected information.

AI Impact on Data:

Elaboration: While the text acknowledges potential AI impacts, it lacks a thorough exploration of how AI within Microsoft products might influence employee behavior and, consequently, the study's outcomes.

Need for Experimentation: Conduct controlled experiments to assess how AI features within Microsoft products impact employee interactions with the intranet. Isolate AI influences to understand their implications on data interpretation and potentially modify the study design accordingly.

Methodological Limitations:

Elaboration: The pragmatic interpretative ideology introduces potential biases, and the text lacks a detailed justification for specific methodological choices, impacting the study's overall rigor.

Need for Experimentation: Experiment with hybrid methodologies that balance flexibility and rigor. Conduct sensitivity analyses to evaluate the impact of varying methodological choices on study outcomes and refine the approach accordingly.

Ethical Considerations:

Elaboration: While the text mentions obtaining ethical clearance, it lacks a comprehensive discussion of specific ethical considerations, especially regarding employee data privacy.

Need for Experimentation: Experiment with more robust ethical guidelines and protocols. Consider piloting surveys or interviews to gauge employee perceptions of data privacy and experiment with measures to enhance informed consent processes.

Inadequate Validation of ML Techniques:

Elaboration: Acknowledging programming skill limitations and the absence of thorough validation of ML techniques raise concerns about the study's reliability.

Need for Experimentation: Collaborate with experienced ML practitioners to review and validate the ML scripts. Conduct additional experiments to assess the robustness of chosen ML techniques, potentially comparing them with alternative methods to ensure the reliability of results.

Lack of Comparative Analysis:

Elaboration: The study identifies personality factors associated with cybercrime risk but does not provide a comparative analysis with existing cybersecurity frameworks or models.

Need for Experimentation: Conduct experiments by incorporating a comparative analysis with established cybersecurity frameworks. Apply ML techniques to datasets used in previous studies to evaluate consistency and identify unique insights the current research provides.

Incomplete Discussion of Results:

Elaboration: The text briefly mentions findings but lacks a comprehensive discussion, limiting the depth of result analysis.

Need for Experimentation: Experiment with more detailed result analyses, potentially leveraging additional statistical techniques. Conduct experiments to explore the practical implications of findings for organizational cybersecurity practices and policies.

Comments on the Quality of English Language

Minor editing of English language required

Author Response

Response to Reviewer 4 Comments

Summary  

Shout out and thank you REVIEWER for taking the time to review this manuscript and for your effort in writing out the constructive improvement suggestions. It is recognized how much of your time this must have taken and rest assured your ideas are now incorporated into the revised version of this paper. Consequently, please see the detailed responses to your constructive suggestions.

As well, the corresponding revisions/corrections have been highlighted in the revised manuscript using the track changes feature (shows up in red). The revision process followed the MDPI recommendations, which were stated as "the compare function in Word can add tracked changes to the final version by comparing it with an earlier version." (https://www.mdpi.com/authors/layout#_bookmark11).  The changes were too lengthy to paste in here so please see the revised manuscript to determine if it complies with your requested revisions and if not please feel free to specifiy additional changes that are need.

Below are the enumerated point-by-point response(s) to your comments and suggestions.  

I would not like to sign my review report - as mentioned by the reviewer.   That is ok. Thank you for these helpful suggestions.   

We can point out right up front that the required BDCC structure has been followed by utilizing the BDCC template (explained below). If there is another required format it would be helpful if the Editor could provide it with directions on its use.  

The introduction has been modified to emphasize the problem statement and research aims. The introduction succinctly introduces the topic in a scholarly manner by grounding the scientific problem/subject into the extant literature. The research objective, goals and research question were stated.  

The terminology used in the article should has been explained clearly, especially cybercrime risk, using all the relevant terms, and personality traits by referencing the a priori literature on the Big Five Personality THeory which is also called Five Factor Personality Theory (abbreviated FFPT, as also explained in the manuscript). The words/phrases were linked to the correct corresponding theory.  Validation has already been accomplished for the FFPT model which is why an a priori theory was selected. The research does not aim to validate or fit the data to the FFPT but rather the FFPT is used as scaffolding input into ML. 

 The introduction has succinctly introduced the topic and provided relevant background information. The introduction clearly states the problem the research aims to solve, it outlines the objectives or goals of the research, and it presents the main argument that the paper will aim to prove. The introduction sets the tone for the rest of the article. The use of citations to the high cybercrime rate was intended to engage the reader’s interest. 

The repetition in the introduction about self-reported research and insiders vs. groups was removed. A more concise and clear presentation was provided.  The reviewer has provided an outstanding succinct statement describing the purpose and this was adapted and then integrated into the paper.  

Ethical Considerations:   Elaboration: While the text mentions obtaining ethical clearance, it lacks a comprehensive discussion of specific ethical considerations, especially regarding employee data privacy.  

Need for Experimentation: Experiment with more robust ethical guidelines and protocols. Consider piloting surveys or interviews to gauge employee perceptions of data privacy and experiment with measures to enhance informed consent processes.   Thank you for this information. The ethical clearance was obtained from the client company as it was an action research project as explained in the methods section. There were no human subjects.

The case study company provided the data using an application programming interphase (API) which the primary investigator advised how to build. The API was run every evening for a month in 2023, to collect data typed into the intranet, but without the identifying end user (employee) identifying attributes. If the data contained any names of any employee in the directory, those names were removed. The resulting big data consisted of only text phrases with no possible identifying information! This meets all research protocols. This API was reviewed by all appropriate case study company staff and approved. In essence, good scientific research requires good scientific data! The company approved the data and the data were used for the study. There is and was no informed consent required. This was an action research study - please refer to action research protocols for more information.  

In addition to the above process, the ethical clearance signed by the case study company project sponsor/authority was provided to the Assistant Editor of the BDCC journal, and it was noted in the comments at the end of the study.  

As explained in previous responses, the "Results: Research findings" and "Discussion" sections were enhanced to ensure there was a correct interpretation of the findings and their implications. In other words, the findings presented the results and the discussion presented the interpretations - but if there is another structural format please advise us.

Methods.  Sampling and Data Collection Issues:   Elaboration: Sampling data at the end of each day may introduce biases and compromise the representativeness of the dataset. Furthermore, data validity and reliability concerns underscore the need for a more robust approach.  

Need for Experimentation: Experiment with alternative sampling strategies, such as continuous random sampling, to mitigate temporal biases. Implement rigorous data integrity checks and explore encryption methods to ensure the accuracy and reliability of collected information.  

Methodological Limitations:   Elaboration: The pragmatic interpretative ideology introduces potential biases, and the text lacks a detailed justification for specific methodological choices, impacting the study's overall rigor.  

Need for Experimentation: Experiment with hybrid methodologies that balance flexibility and rigor. Conduct sensitivity analyses to evaluate the impact of varying methodological choices on study outcomes and refine the approach accordingly.  

Inadequate Validation of ML Techniques:   Elaboration: Acknowledging programming skill limitations and the absence of thorough validation of ML techniques raise concerns about the study's reliability.  

Need for Experimentation: Collaborate with experienced ML practitioners to review and validate the ML scripts. Conduct additional experiments to assess the robustness of chosen ML techniques, potentially comparing them with alternative methods to ensure the reliability of results.  It was suggested by the review that the methods section should clearly describe the research design and methods used, potentially using diagrams and tables for clarity. This was an excellent idea - so diagrams were developed for this study, and placed into the manuscript with more explanation.  

To address the concern, a flexible approach was used to investigate new or challenging issues, and explained in the methodology - the explanation was included and cited to a Springer Nature/Macmillan textbook of research methods to ensure credibility and to provide other researchers with a grounding point. This explanation will hopefully improve the rigor, thoroughness, consistency, reliability, and validity of the study. The author/s spent considerable effort explaining the research procedure using a diagram to inform future research.   

To address another issue with the methodology, regarding the procedures, these were detailed. The method diagram and accompanying explanation will hopefully achieve this. Other reviewers provided salient advice for how to do that, as described later. The FFPT model factors are a priori and cited to the literature. The revised methodology and procedure explains how the FFPT factors were leveraged in the current study.  

The explanation of how the experiment was run was explained in detail with the aid of a diagram with accompanying explanation. Perhaps some aspects were too detailed, but other researchers will need to know those details and this is necessary to achieve scientific rigor.   

There was validation of the ML techniques through the limited learning and training exercise using the CHAID process! However, we forgot to include the validation estimates, the accuracy scores which we added. Thank you for all the advice.  

Comments and Suggestions for Authors Limited Generalizability:   Elaboration: The study's reliance on data from a single multinational company restricts the generalizability of findings, limiting their applicability across diverse industries and organizational cultures.  

Need for Experimentation: Conduct the study across various companies representing different sectors and geographical locations to enhance external validity. Collaborate with multiple organizations to gather a more comprehensive dataset, allowing for comparative analyses and broader applicability.  

AI Impact on Data:   Elaboration: While the text acknowledges potential AI impacts, it lacks a thorough exploration of how AI within Microsoft products might influence employee behavior and, consequently, the study's outcomes.

Need for Experimentation: Conduct controlled experiments to assess how AI features within Microsoft products impact employee interactions with the intranet. Isolate AI influences to understand their implications on data interpretation and potentially modify the study design accordingly. Thank you for this information. There was no section in the BDCC template named "implications" but it is generally understood that this would be presented in the discussion and or conclusions sections.  

The implications of the findings and the contribution were strengthened in the discussion/conclusions section. The study utilized real-world big data from a company intranet to discover the risk of cybercrime in employee online behavior. The clarity and significance of the contributions were made clear. This was also done in the abstract (both the neuroticism factor findings as well as the novel ML procedure).   

It was acknowledged that this study relied on data from a single multinational company which restricts the generalizability of findings, limiting their applicability across diverse industries and organizational cultures.   It was suggested that future researchers conduct the study across various companies representing different sectors and geographical locations to enhance external validity, as well as to collaborate with multiple organizations to gather a more comprehensive dataset, allowing for comparative analyses and broader applicability.  

It was also pointed out in the discussion/conclusions that while the paper acknowledged the potential AI impacts, it lacked a thorough exploration of how AI within Microsoft products might influence employee behavior and, consequently, the study's outcomes - which was beyond the scope of the current study.  

It was admitted that we need to conduct controlled experiments to assess how AI features within Microsoft products impact employee interactions with the intranet. Isolate AI influences to understand their implications on data interpretation and potentially modify the study design accordingly. Good points. 

Lack of Comparative Analysis:   Elaboration: The study identifies personality factors associated with cybercrime risk but does not provide a comparative analysis with existing cybersecurity frameworks or models.  

Need for Experimentation: Conduct experiments by incorporating a comparative analysis with established cybersecurity frameworks. Apply ML techniques to datasets used in previous studies to evaluate consistency and identify unique insights the current research provides.   Yes this paper is an applied action research project - a scientific method in itself, as described in the literature (refer to Sage or Springer Nature for research methods textbooks).   

The discussion/conclusions summarized the key findings and implications from the study results.  As noted, the paper author(s) PRESENTED AN intriguing finding high neuroticism and low openness to new experiences were associated with employees potentially having a higher cybercrime risk. The methodology used to establish this association was explained as statistical Euclidean distance coefficients associated with word associations between the PPFT, risky URLs, and what employees had typed in the intranet.

The author/s elaborated on the measures used to determine the higher cybercrime risk coefficient in tghe methods section. Furthermore, the implications of this finding, both theoretical and practical, were explicitly stated.

In other words, the manuscript shows how this high cybercrime risk finding contributes to the existing body of knowledge and how it can be applied in a real-world context.   As noted above, the author/s included details about the measures used to determine cybercrime risk in the methods, and the potential applications of those findings were discussed in more depth in the discussion/conclusions section. The PPFT instrument is already well-described in the literature and there are space limitations in the BDCC journal that prevent adding more words to the manuscript.  

Incomplete Discussion of Results:   Elaboration: The text briefly mentions findings but lacks a comprehensive discussion, limiting the depth of result analysis.  

Need for Experimentation: Experiment with more detailed result analyses, potentially leveraging additional statistical techniques. Conduct experiments to explore the practical implications of findings for organizational cybersecurity practices and policies.      

Thank you. The paper has been revised to discuss the findings with a comprehensive discussion of the result analysis.   Certainly there is a need for Experimentation: It was suggested that future researchers experiment with more detailed result analyses, potentially leveraging additional statistical techniques. Conduct experiments to explore the practical implications of findings for organizational cybersecurity practices and policies.  

Comments on the Quality of English Language Minor editing of English language required

The author(s) were born in the USA and they are fluent in the English language as well as in more than one language. However, typos are inevitable with large manuscripts, so the manuscript was carefully read by the primary investigator and three doctoral students in American-accredited universities to correct any grammar errors. The three doctoral students were paid $1000 each to ensure the grammar was correct (which was deemed to be a fair price to achieve the quality necessary for the BDCC journal).

The author(s) are open and willing to make other required changes identified by the reviewer(s) or editor(s).  The title and abstract have been modified to be in line with above, as quoted below:   "Discovering Cybercrime Risk in Employee Behavior Big Data with Unsupervised Machine Learning and Personality Theories"   "A critical global problem is that ransomware cyber attacks can be costly to organizations. Moreover, accidental insider cybercrime risk can be challenging to prevent even by leveraging advanced computer science knowledge.

This exploratory project used a novel cognitive computing design featuring explanations of the action-research case-study methodology, details of the customized machine learning (ML) techniques with a workflow diagram, and a proposed future research design to inspire scholars. The author analyzed over 8 GB of big data on employee behavior from a Fintech company intranet and then integrated psychology discipline theories into semi-supervised ML to identify cybercrime risk. The ML techniques included preprocessing, normalization, tokenization, sentiment analytics, CHAID learning tree analysis, credibility/reliability/validity checks, heatmaps, and scatter plots of the employee behavior using business language dictionaries enhanced with the five-factor personality theory.

Higher levels of employee neuroticism were associated with a greater organizational cybercrime risk, corroborating the findings in empirical publications. In stark contrast to the literature, an openness to new experiences was inversely related to cybercrime risk. Other employee personality behavior factors had no informative association with cybercrime risk.

This study introduced an interdisciplinary paradigm for big data cognitive computing by illustrating how to integrate the best of proven theories from the literature with ML to analyze human behavior using a retrospective big data collection approach that may sometimes be more efficient, reliable, and valid as compared to traditional methods like surveys or interviews."  

The other changes are too lengthy to paste in, so please kindly see the manuscript.   Again - thank you REVIEWER for taking the time to review this manuscript!

Round 2

Reviewer 1 Report

Comments and Suggestions for Authors

See attachment

Comments on the Quality of English Language

I have identified the problems with the writing style in the attachment.  The tone is informal at times, and the writing style is cumbersome - at times, difficult to read and understand.

Author Response

Thank you so much. I tried my best to apply all the requested revisions, and I ran it through Gammarly with a 100% result, at least for language punctuation and so forth.

I attached my responses into your PDF. I prepared the revised version to hopefully meet your requirements.

Reviewer 4 Report

Comments and Suggestions for Authors

The authors' response was deemed unsatisfactory, as the paper lacked sufficient experimentation for publication. Despite requesting revisions to include experimentation, the authors indicated that "such enhancements would be considered as future work". The current state of the paper renders it unsuitable for acceptance. Therefore, a strong rejection is recommended.

Author Response

Thank you again for re-reviewing this. I am happy to let you know that we will agree to conduct more experimentation or experiments for you as long as you are able to fund the extension to this project (please share the details with the editor so your contact information can be passed to us). We estimate it would take another 4-5 months to complete your requirements. The current paper should be adequate in terms of meeting the scope of the RQ and we would like to submit the revised version.

There is no experimentation in the paper since it's a case study not an experiment now.

Again thank you for you efforts. Much appreciated.

Round 3

Reviewer 4 Report

Comments and Suggestions for Authors

While there have been improvements in the paper, my satisfaction with the experimental section remains incomplete, and as a result, my acceptance of this article remains on the borderline.

Author Response

Response to review 4
